# Towards Efficient and Scalable Implementation of Differentially Private Deep Learning

## Abstract

Differentially private stochastic gradient descent (DP-SGD) is the standard algorithm for training machine learning models under differential privacy (DP). The most common DP-SGD privacy accountants rely on Poisson subsampling for ensuring the theoretical DP guarantees. Implementing computationally efficient DP-SGD with Poisson subsampling is not trivial, which leads to many implementations ignoring this requirement. We conduct a comprehensive empirical study to quantify the computational cost of training deep learning models under DP given the requirement of Poisson subsampling, by re-implementing efficient methods using Poisson subsampling and benchmarking them. We find that using the naive implementation DP-SGD with Opacus in PyTorch has between 2.6 and 8 times lower throughput of processed training examples per second than SGD. However, efficient gradient clipping implementations with e.g. Ghost Clipping can roughly halve this cost. We propose alternative computationally efficient ways of implementing DP-SGD with JAX that are using Poisson subsampling and achieve only around 1.2 times lower throughput than SGD based on PyTorch. We highlight important implementation considerations with JAX. Finally, we study the scaling behaviour using up to 80 GPUs and find that DP-SGD scales better than SGD.

## 1 Introduction

Training data of machine learning (ML) models can be vulnerable to extraction (Balle et al., 2022; Carlini et al., 2021). Differential Privacy (DP) (Dwork et al., 2006) is the gold standard for formalizing the privacy leakage of training data in ML and mitigating the risk of privacy attacks on the training data. DP is deployed in many applications involving sensitive data (Abowd, 2018; Cormode et al., 2018).

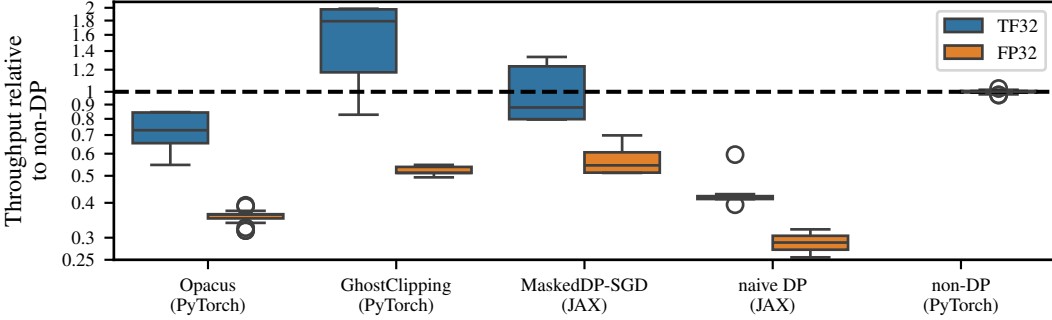

Figure 1: Relative throughput to the respective non private baseline (higher is better) on NVIDIA A100. For each optimization method and each model size, we divide its throughput with the non-private counterpart. Throughput is the number of processed instances per second. We distinguish between precision modes. They are available on both frameworks and significantly improve the throughput for the different DP-SGD implementations.

The established algorithm for integrating DP into the training pipeline of deep learning models is DP stochastic gradient descent (DP-SGD) (Rajkumar & Agarwal, 2012; Song et al., 2013; Abadi et al., 2016), which is the DP adaptation of SGD (see also Algorithm 1). DP-SGD has two major

drawbacks in comparison to SGD: higher computational cost and loss in utility. DP-SGD requires more memory and is computationally more expensive due to the per-example clipping. The utility in comparison to non-DP training drops but this can be mitigated to certain extend by using larger batch sizes (Räisä et al., 2024) and training longer (Ponomareva et al., 2023) which further increases the computational cost.

Standard DP privacy accountants assume so-called Poisson subsampling, where each example is selected at each iteration independently with a fixed probability. This implies that different mini-batches will be of different size, and makes efficient implementation more difficult. As a result, many existing implementations forego proper implementation of Poisson subsampling. Recent research (Lebeda et al., 2024; Chua et al., 2024a;b; Annamalai et al., 2024) shows that such implementations may have significantly weaker privacy guarantees than claimed under the Poisson subsampling assumption and that Poisson subsampling is essential for achieving optimal utility under DP.

**List of contributions** In this work we perform an extensive empirical study on the computational efficiency of DP-SGD. We will focus on fine-tuning a wide range of large computer vision classification models but our findings can be applied any other large models that are trained or fine-tuned with DP-SGD. Our main contributions are the following:

1. We re-implement all DP-SGD methods with Poisson subsampling that is fully DP and share the source code.
2. We find that non-optimized training with DP-SGD costs per-epoch between 2.6 and 3.2 times more than non-private training for ViT and 4 to 8 times for ResNets (See Section 4). We identify the reasons that lead to the higher computational cost of DP-SGD using profiling.
3. We benchmark different strategies that can reduce this cost drastically up to a level that matches the non-optimized non-private training (See Figure 1 for an overview): (i) More efficient gradient clipping implementations of DP-SGD (See Section 5.1). (ii) Lower Precision with TF32 (See Section 5.2).
4. We propose a JAX implementation relying on proper Poisson sampling that is not prone to re-compilation and outperforms a naive implementation in terms of throughput (See Section 6).
5. We scale up the training to 80 GPUs and find that DP-SGD scales better than non-private training (See Section 7).

## 2 BACKGROUND

This section will explain the main DP-SGD algorithm and optimizations to alleviate its computational cost.

### 2.1 DP-SGD ALGORITHM

Algorithm 1 is the original DP-SGD algorithm, with virtual batching, as proposed by Abadi et al. (2016).

**Virtual Batching** distinguishes between logical and physical batches. Logical batches are divided into multiple physical batches to allow taking optimizer steps with many samples without running out of memory. For example, we typically sample logical batch sizes of size $L = 25000$ while the memory only fits $< 300$ samples at a time. Implementing DP-SGD with virtual batching Algorithm 1 does not modify the privacy accounting. The amount of added noise is the same and does not affect the model utility (Ponomareva et al., 2023).

**Poisson subsampling** Interestingly, Bu et al. (2022) and Bu et al. (2023) never mention Poisson subsampling in their works of Mix Ghost clipping and Book Keeping. Even more, Bu et al. (2022) state that it has a speed-up of $\times 1.7$ times against other algorithms with a fixed batch size, which would affect the privacy accounting method. The same happens in practice for JAX implementations (De et al., 2022), where the sampling is done by shuffling the dataset and using each sample once per epoch. While this makes efficient implementation easier, it does not use the correct Poisson subsampling assumed by the privacy accounting methods, and therefore the implementation might have significantly weaker privacy properties than claimed (Lebeda et al., 2024; Chua et al., 2024a;b; Annamalai et al., 2024). All our experiments are based on Poisson subsampling which is compliant with the commonly used privacy accounting.

---

**Algorithm 1** Virtual Batching DP-SGD

---

**Input:** Training data points $\{x_1, \ldots, x_N\}$, loss function $\mathcal{L}(\theta) = \frac{1}{N} \sum_i \mathcal{L}(\theta, x_i)$
**Parameters:** Parameters: learning rate $\eta_t$, noise scale $\sigma$, gradient norm bound $C$, number of steps $T$, approximate logical batch size $L$, physical batch size $p$.
**for** $t \in [T]$ **do**
    $B \leftarrow \{x_{j_1}, \ldots, x_{j_m}\}$ with Poisson sampling with rate $L/N$.
    $P \leftarrow \{B_1, \ldots, B_k\}$ divide the logical batch $B$ into physical batches of size $p$.
    $\theta_{acc} \leftarrow \mathbf{0}$
    **for** $s \in [P]$ **do**
        For each $i \in s$ compute $\mathbf{g}_t(x_i) \leftarrow \nabla_{\theta_t} \mathcal{L}(\theta_t, x_i)$ {**Compute gradient**}
        $\overline{\mathbf{g}}_t(x_i) \leftarrow \mathbf{g}_t / \max(1, \frac{\|\mathbf{g}_t(x_i)\|_2}{C})$ {**Clip gradient**}
        $\theta_{acc} \leftarrow \theta_{acc} + \sum_i \overline{\mathbf{g}}_t(x_i)$ {**Accumulate gradient**}
    **end for**
    $\widetilde{\mathbf{g}}_t \leftarrow \frac{1}{|L|}(\theta_{acc} + \mathcal{N}(0, \sigma^2 C^2 \mathbf{I}))$ {**Add noise**}
    $\theta_{t+1} \leftarrow \theta_t - \eta_t \widetilde{\mathbf{g}}_t$ {**Step**}
**end for**
**Return** Learned parameters $\theta_T$ and the privacy cost from a privacy accountant.

---

## 2.2 DP-SGD Gradient Clipping Optimizations

We benchmark five types of clipping methods. Table 1 shows which clipping optimizations we are benchmark against the library or framework that implements it.

Table 1: Benchmarked DP-SGD frameworks and libraries. Note that Opacus implements Ghost Clipping but in our experiments the loss does not decrease, thus indicating a problem.

| | | PyTorch | | | JAX | |
| Clipping Mode | Native | Opacus (Yousefpour et al., 2021) | PrivateVision (PV) (Bu et al., 2022) | FastDP (BK) (Bu et al., 2023) | Native | Ours |
|---|---|---|---|---|---|---|
| Non-private | ✓ | | | | ✓ | |
| Per-example | | ✓ | | | ✓ | |
| Ghost Clipping (Li et al., 2022) | | ? | ✓ | ✓ | | |
| Mix Ghost (Bu et al., 2022) | | | ✓ | ✓ | | |
| Mix Opt (Bu et al., 2023) | | | | ✓ | | |
| Masked DP-SGD | | | | | | ✓ |

**Ghost clipping** computes the loss gradient norm after the backpropagation optimization and then reweights the loss to update the clipped gradients. Its main contribution is memory saving at the cost of adding another backward pass (Li et al., 2022).

**Mixed Ghost clipping** (Bu et al., 2022) builds on-top of Ghost clipping. It implements the ghost clipping technique for convolutional layers. Its main contribution is that the algorithm will decide when to clip the gradients using per-example or ghost. This difference matters because the ghost clipping is less efficient when the layer's input dimensions are too big. E.g., for ResNets, each clipping method will be applied for half of the layers. The first layers will be clipped using the the per-example and then ghost clipping in the bottom layers. As the model goes deeper, the feature size decreases, and the number of channels increases, prioritizing ghost clipping (Bu et al., 2022).

**Book Keeping** (Bu et al., 2023) uses all the previous techniques but requires only one backpropagation pass without explicitly calculating the per-example gradients. It avoids the second pass that ghost clipping does by reusing the intermediate results of the output gradients to calculate the sum of the clipped gradients and the clipping factor. Book Keeping can also be implemented together with the mix optimization. It does the same evaluation as the mix ghost clipping, but also determines whether doing a second backward pass is more efficient.

## 3 Experiment Overview

**PyTorch implementations** We benchmark a native PyTorch (Ansel et al., 2024) implementation with PyTorch based libraries Opacus (Yousefpour et al., 2021) (details on gradsampling in Appendix A.3), PrivateVision (PV) (Bu et al., 2022), and FastDP (BK) (Bu et al., 2023), see Table 1.

At submission time ghost clipping in Opacus was still undergoing changes and was unstable in our experiments.

**Native JAX implementation** We benchmark a native JAX (Bradbury et al., 2018) implementation that clips the per-sample gradients with Optax (DeepMind et al., 2020) without utilizing any further optimization. This naive implementation in JAX is prone to recompiling due to changing tensor sizes caused by the Poisson subsampling.

**Masked DP-SGD** We propose an alternative algorithm called Masked DP-SGD which solves the issue of recompilation but computes always slightly more gradients. It consists of the following sub-steps at every iteration (see Algorithm 2 for more details):

1. We sample a logical batch size using Poisson sampling.
2. We round up the number of samples for which we compute per-sample gradients so that it is devisable by the physical batch size without remainder.
3. We compute the per-sample gradients.
4. We mask out gradients so that the per-sample gradients used for the update are the actual Poisson subsampled ones, ensuring compliance with the Poisson subsampling accounting.

**Implementation of Poisson sampling** Opacus samples the logical batches using Poisson sampling and then divides them into physical batches using their `BatchMemoryManager` class. The other implementations considered in our experiments do not support virtual batching out-of-the-box. To make a fair comparison between all methods, we implement the Poisson subsampling, the same way Opacus does it, for all frameworks and adapt the Batch Memory Manager to support them. This way, all the experiments are seeded and have the same logical batch sizes.

**Metrics** We compare the throughput, defined as how many samples can be processed per second during training, and the maximum physical batch size, reached before running out of memory.

**Dataset** We benchmark with the CIFAR100 (Krizhevsky & Hinton, 2009) resized to 224x224.

**Models** We train two families of models: Vision Transformer (ViT) (Dosovitskiy et al., 2021) and ResNet (Kolesnikov et al., 2020) (See Table 2). Both are pre-trained on ImageNet-21k (Russakovsky et al., 2015).

Table 2: Number of parameters, in millions, for each family architecture and size of the model.

| VISION TRANSFORMER (ViT) | | RESNET | |
|---|---|---|---|
| TYPE | # OF PARAMETERS | TYPE | # OF PARAMETERS |
| TINY | 5.7 M | $50 \times 1$ | 23.7 M |
| SMALL | 22.1 M | $101 \times 1$ | 42.7 M |
| BASE | 86.6 M | $50 \times 3$ | 211.8 M |
| LARGE | 304.3 M | $101 \times 3$ | 382.4 M |
| HUGE | 630.8 M | $152 \times 4$ | 929.2 M |

**Parameterization** While parameter-efficient fine-tuning of some parts of the model has been shown to be effective under DP (Tobaben et al., 2023; Yu et al., 2022), our work focuses on the computational efficiency of DP-SGD and thus we consider the worst-case scenario of fine-tuning all parameters of the model. Furthermore, any training from scratch requires training all parameters.

**Hyperparameters** We train for four optimizer steps with a sampling rate of $0.5$ (expected batch size of 25000), which allows us to test the experiments quickly with a realistic high batch size (Ponomareva et al., 2023; Räisä et al., 2024). We do not focus on finding the best possible utility, which requires training for many more epochs (See Table A2 for the accuracy after training for four steps).

**Environment specifications** We use two GPU architectures: NVIDIA V100 (32 GB VRAM) and A100 (40 GB VRAM) with identical Python environments. Each node contains four GPUs. We use 16 CPU workers for data loading. In the distributed case of more than one GPU, we cannot use multiple workers.

**Source code** We provide the code for reproducing the experiments in the supplementary material and will publish the code in an open repository after acceptance of the paper.

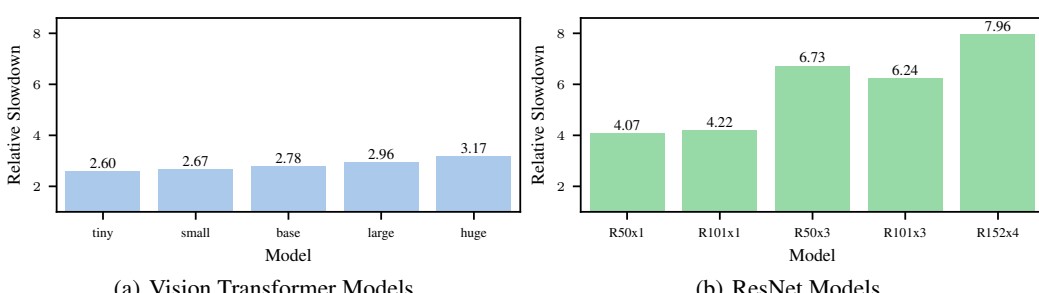

(a) Vision Transformer Models        (b) ResNet Models

Figure 2: Relative slowdown in mean throughputs between Opacus per-example clipping and the non-private baseline using one A100 GPU. It is defined as private-throughput/non-private-throughput, this means that lower is better. It shows how many times private training is more expensive with a relative slowdown of 1 indicating that Opacus is as fast as non-private training.

## 4 WHAT IS THE COST OF DP IN DEEP LEARNING?

In this section we will quantify the computational cost of deploying DP training. We do this by comparing the throughputs and maximum physical batch sizes between the non-private training with PyTorch and private training with Opacus, which is the most used DP-SGD implementation. Additionally, we identify the reasons for the higher computational cost of DP-SGD through profiling.

### 4.1 THROUGHPUT AND MAXIMUM BATCH SIZE COMPARISON

We compare relative throughput (Figure 2) and the maximum physical batch size (Figure 3) between DP-SGD (Opacus) and non-private training with PyTorch. The main metric of interest is the throughput as it quantifies the training speed but the maximum physical batch size becomes important when training models that are too large to fit even one example at a time. For both metrics DP-SGD becomes more expensive with larger models but the detailed trends differ.

**Vision Transformer** The throughput difference between Opacus and the non-private baseline with PyTorch (see Figure 2(a)) does not spike but grows steadily as a function of model size, which is interesting considering how big the relative difference is in maximum physical batch size (Figure 3(a)) is: The throughput ranges from a relative difference of $\times 2.6$ for the smallest model to $\times 3.17$ for the largest model while the maximum physical batch size has a relative difference of around $\times 4$ for the smallest model and $\times 11$ for the largest model.

**ResNets** We observe a less steady growth trend in terms of throughput with the ResNets in Figure 2(b). When increasing the model size the ResNets do have spikes of growth as the model size grows. The contrast in Figure 2 between ViT and BiT ResNet models is due to the architecture and types of layers. The parameter space grows as the width factor (see Table 2) for the ResNets, so the $\times 3$ makes the neural network wider by a factor of three. Based on our results, the width of the layers affects throughput much more than the depth of the network. They have comparable throughput with the same width and different depths, but increasing the width will make the model in the private setting much slower and reduce the maximum batch size significantly.

**How much does finding the maximum physical batch size matter?** In Figure A.1 in the Appendix, we display the relative throughput as a percentage by dividing the throughput at a particular physical batch size by the maximum achievable throughput. We see that as the physical batch size increases, the throughput will grow as expected, but at some point there is no significant further improvement in throughput from using a larger physical batch size. Practitioners may estimate the optimal batch size based on available memory and performance trade-offs. It is not crucial to set the physical batch size to the maximum possible but a good enough value is fine. Typically, throughput using smaller batches is limited by data loading speeds, but as batch size increases, computation becomes the limiting factor.

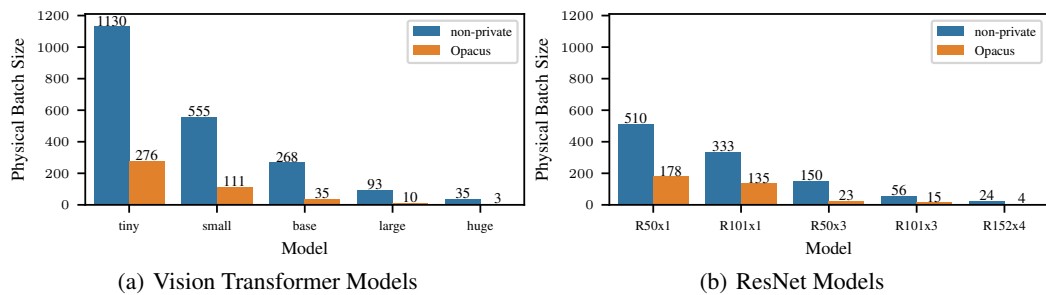

(a) Vision Transformer Models  (b) ResNet Models

Figure 3: Maximum achievable physical batch size by the different model sizes on A100 GPU (40 GB) before they reach Out Of Memory (OOM) Error. The model sizes grow from left to right. To check the number of parameters of each size, refer to Table 2.

## 4.2 Reasons for the Increase in Computational Cost

Giving a detailed breakdown of low-level operations associated with DP is challenging. However, using GPU profiling tool NVIDIA Nsight System, we can identify three aspects which constitute the majority of DP overheads. Firstly, due to its larger memory footprint, DP-SGD is able to consume smaller physical batches than its non-private counterpart. This results in larger amount of smaller low-level kernel calls which leads to slightly lower utilisation of the GPU. At very small batch sizes even the kernel launch overheads can become a notable factor for slowdown. Secondly, the computation of per-example gradients introduces significant overhead in the backward pass as it cannot be parallelised as in batched gradient computation. This is the most prominent cause of the total overhead. Finally, an additional DP-optimizer step that clips and accumulates the per example gradients, which is not present in the non-DP algorithm, needs to taken after each physical batch (see Table 3).

Table 3: Average processing time for each section of the algorithm. We are comparing the non-private and Opacus per-example clipping on A100, with the same physical batch size. It is calculated with NVIDIA Nsight Systems. All the measurements include the syncronization time, which is needed for the profiling, but adds additional time that is not part of the normal execution. All values are in milliseconds.

| Section | PyTorch non-private | Opacus per-example |
|---|---|---|
| Forward | 81.14 | 101.53 |
| Backward | 163.85 | 681.48 |
| Clip and Accumulation | 0 | 26.76 |
| Optimizer Step | 38.17 | 99.65 |

## 5 Decreasing the computational cost

This section analyzes the different strategies for training with DP-SGD more efficiently. We evaluate both algorithmic and hardware optimizations and their combinations.

### 5.1 Efficient gradient clipping algorithms

First, we evaluate the more efficient gradient clipping implementations that have been described in Section 2.2 using the Vision Transformer ViT base model. We chose it as our benchmark model because the middle model size is large enough to evaluate the advantages of the optimized gradient clipping algorithms but does not require excessive amount of time to train. The non-Opacus implementations do not support the BiT ResNet due to their custom weight standardization layer.

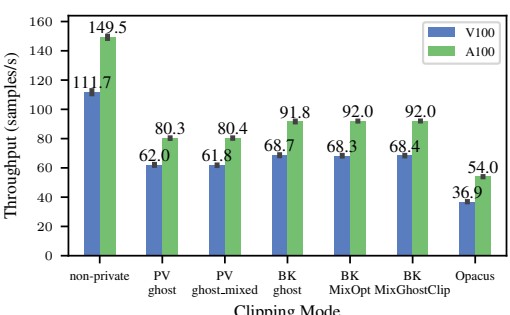

Figure 4: Throughput using the maximum batch size for each clipping algorithm. It compares the executions for both V100 and A100, for the ViT Base model.

Table 4: Maximum physical batch size reachable for each clipping method, for the two GPU architectures we are comparing, for the ViT base model.

| CLIPPING MODE | V100 (32GB) | A100 (40GB) |
|---|---|---|
| NON PRIVATE BASELINE | 216 | 268 |
| PER-EXAMPLE (OPACUS) | 28 | 35 |
| GHOST (PRIVATE VISION) | 203 | 257 |
| MIX GHOST (PRIVATE VISION) | 203 | 257 |
| BK GHOST (FASTDP) | 189 | 209 |
| BK MIX GHOST (FASTDP) | 189 | 209 |
| BK MIX OPT (FASTDP) | 189 | 209 |

**Throughput Comparison** Figure 4 displays the throughput for each clipping algorithm for each tested GPU. Moving from a V100 to an A100 GPU increased the throughput by $\times 1.3$ times on average over all clipping methods. The one that benefited the most is the per-example clipping by Opacus with a $\times 1.46$ improvement in throughput. This is because of Opacus-specific optimizations. Their implementation is optimized to vectorize the virtual batches and get the most out of the processing unit to compensate for the per-example clipping. Also, we base our virtual batching module on Opacus, which may have further contributed to the advantage seen for Opacus. The other implementations showed benefits similar to those of non-private training. In both GPU architectures, the clipping optimizations consistently maintained their relative throughput difference to their non private baseline. Private Vision gets closer to the non-private baseline physical batch size, but Book Keeping is closer to its throughput with a smaller physical batch size (see Figure 6).

Without sacrificing utility (Table A2), these optimizations offer an alternative to the original per-example clipping algorithm. Even though Book Keeping performs better in throughput, it is by a very narrow margin. Consequently, Private Vision and FastDP remain viable options for implementing ghost clipping. The difference between the two algorithms is the second backward pass over the neural network. Since the Book Keeping trick avoids doing the second backward pass through the network, it has a higher throughput at a small memory cost.

Mixing ghost clipping does not yield any improvement because it determines whether it should apply ghost or per-example clipping, which depends on the size of the inputs and the parameter space. If the dimensions are large enough, the ghost technique will be more expensive (Bu et al., 2022). In a ViT model, the dimensions change less than in a convolutional network. Therefore, despite continually evaluating which method to apply, it always uses ghost clipping. However, if applied to a ResNet model, it should outperform ghost clipping, as it is optimized for convolutional layers. It could not be tested on BiT ResNet models used in this study due to incompatibilities with the Private Vision and FastDP, preventing an assessment of mixed optimization methods.

**Maximum physical batch size** Table 4 compares the maximum physical batch size for both available GPUs. The maximum physical batch size is larger for the optimizations of DP-SGD than for Opacus because they do not require per-example gradients. Thus, the optimizations allow training much larger models without running out of memory. The maximum physical batch size using Private Vision library is the one that comes closest to the non-private baseline. In general, we can see that the methods are consistent within implementations, with Private Vision and the FastDP reaching the same maximum physical batch size no matter the clipping mode. As expected, the A100 achieves consistently higher maximum physical batch sizes than the V100 due to the larger amount of VRAM.

### 5.2 LOWER PRECISION

We consider using lower precision to speed up computation. We evaluate the use of Tensor Core 32 (TF32) for training. TF32 has 10 bits for precision, with eight range bits, giving it the same range but less precision than 32-bit single precision floats (FP32) (Kharya, 2020). Using lower precision can have benefits exactly where DP training struggles: it requires less memory, uses fewer bits to

represent the data, and its operations are optimized for GPU, making them much faster (NVIDIA, 2023). It is specially optimized for the A100 GPU and unavailable for the V100, so we compared training on the A100 with and without TF32.

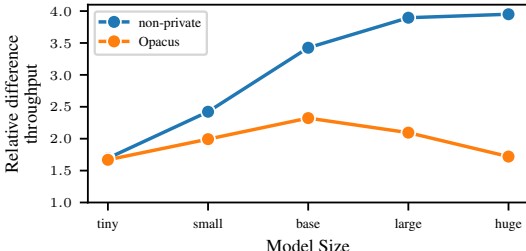

Figure 5: Relative difference in mean throughput between TF32 and FP32 Training for ViT Models.

**Experimental results** In Figure 5, we display the relative difference between mean throughput between runs with TF32 and FP32. For non-private training, throughput increases with model size. For private training throughput increases for the smaller models, but it goes down again as the model size grows after the base size. Models that are too small do not gain much from TF32, and the larger ones have too little maximum physical batch size to benefit (See detailed discussion of this in Appendix C). Regarding the memory advantages by TF32, we could not see an improvement. Both models, with and without TF32, could fit the same number of instances.

**Concerns regarding TF32 under DP** There are two concerns with using lower precision in DP deep learning: its effects on utility and privacy. For the first issue, using lower precision may affect utility, as it is less precise. We did not find a significant decay in the accuracy of the models compared to the models with FP32; it differs by decimal points at the $\times 10^{-6}$ precision (See Table A2 in the Appendix). Regarding privacy, all floating point implementations provide imperfect implementations of real-valued mechanisms, and this can cause additional privacy vulnerabilities (Mironov, 2012). Using lower precision may exacerbate the problem. Discrete mechanisms (e.g. Canonne et al., 2020; Agarwal et al., 2021) that avoid the theoretical challenges exist, but they are often less convenient and may lead to loss of utility, especially in low precision settings. The efficiency of different discrete mechanisms in TF32 is an interesting topic of further research.

## 6 JAX

In this section we compare the performance of the two JAX implementations with all other DP-SGD frameworks (all of them are based on PyTorch). The utility is the same as in PyTorch, see Table A2.

**Compilation time** Comparing JAX to PyTorch requires taking the compilation time into account that the DP-SGD implementations in PyTorch do not utilize. There is no straightforward way of calculating the compilation time, but we measure it as the duration to process the first batch. The execution times for each batch shows that the first one takes much more time than the others, which means it includes the compilation time (Figure A.2). To provide a fair comparison, we also implemented a non-private JAX training using the same virtual batching as PyTorch.

**Throughput comparison** JAX defaults to TF32 when it is available. Therefore, to compare it with Torch, we forced the use of higher-precision FP32. In Figure 6, we compare both precision modes. A naive JAX implementation is as slow as Opacus due to JAX recompilation. When JAX defaults to TF32, our Masked implementation outperforms all methods and is steady, as the batch size does not affect the throughput as much. But we can also see how TF32 benefits Torch implementations, which are much more dependent on the batch size. At the largest physical batch size, Opacus with TF32 can have a higher throughput than our method (See Figure 6(b)).

The masked DP-SGD jax implementation shows a higher throughput than the other JAX implementations. Since for this implementation we are fitting the whole logical batch in CPU memory, we can split it and have static sizes. Therefore the compilation time will be higher for the first logical batch, but for the next iterations we can see the gains in speed, as it does not need to recompile. Its throughput changes less with respect to its batch size, in comparison to other methods. See Appendix D for a comparison to a concurrent method by Chua et al. (2024b).

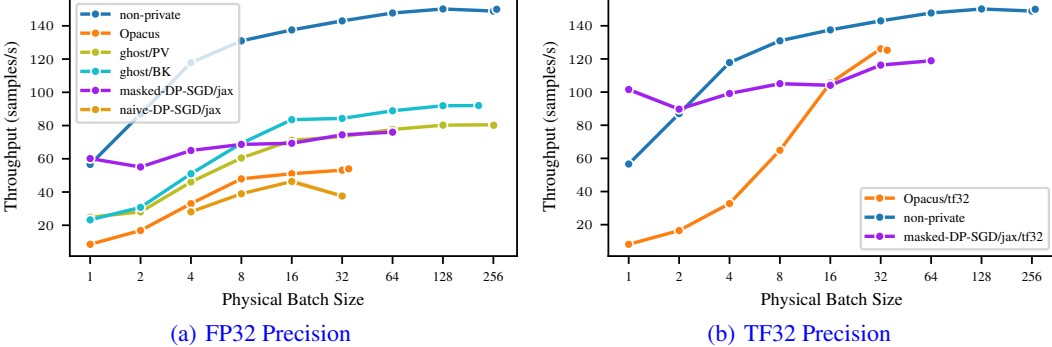

(a) FP32 Precision  (b) TF32 Precision

Figure 6: Comparison of the throughput as a function of the physical batch size between the JAX and PyTorch clipping algorithms on A100 GPU. Only the ghost implementations from Private Vision and Book Keeping are used, not the Mix algorithms, since they have the same performance. The estimator is the median, and the error bars are the $95\%$ confidence interval using bootstrapping. (a) Throughput for all methods with FP32 precision. (b) Comparison between Opacus and our Masked DP-SGD implementation with TF32 precision.

Our Masked implementation of per-example clipping reaches a higher physical batch size, and the throughput is always higher than its Opacus counterpart in the FP32 setting. It is as efficient in terms of throughput as the Private Vision Ghost Clipping. However, the Book Keeping Ghost Clipping implementation has a higher throughput after a physical batch size of 8. For practitioners that remain in PyTorch, Book Keeping ghost clipping presents a throughput comparable to the execution of the Masked DP-SGD JAX while reaching a larger batch size.

Another difference between the two frameworks is the variability in the experiments. PyTorch runs are remarkably consistent in having a low variance, and the same throughput result is expected every time for a fixed seed. JAX executions are more variable than those of PyTorch, likely due to its sensitivity to HPC environment fluctuations and accelerator stochasticity, as noted in Figure A.2. Another contributing factor is JAX's asynchronous dispatch method, which complicates time benchmarking by issuing a promise rather than immediate results, concealing Python overheads.

The compilation time (see Figure A.2) grows with the batch size. For the private model, it takes more time since the compiled function is more complex than the non-private counterpart. It includes expanding the dimensions and clipping the gradients, while the non-private directly computes the gradient of the whole mini-batch.

**Poisson sampling** Using JAX for DP introduces complexities, particularly around subsampling which is crucial for privacy accounting. Implementing Poisson subsampling results in variable batch sizes in JAX; changes in batch size require JIT to recompile, leading to graph retracing which is costly and contributes to execution run variability, as discussed by Chua et al. (2024a). Our masked DP-SGD implementation overcomes this issue while using proper Poisson subsampling and therefore ensuring the correct privacy budget.

**PyTorch Compilation** Although compiling PyTorch is possible, we could not see any improvements in terms of speed-up. While compiling the non-private model worked, the speed-up gained was minimal and, in the end, even lower if we consider the compilation time. PyTorch also recompiles after a batch size change. While trying to compile, PyTorch falls back to predefined CUDA operations that are already optimized. In the case of the private setting, the compilation does not recognize Opacus hooks and continues the execution without compiling them (See Figure A.3).

Leveraging the same kernels to support the private hooks and avoid the compilation would require massive engineering work of writing special kernels for each specific private case. On the other hand, JAX will compile the JIT functions in XLA, but it does not fall back to the kernels, making it more generalizable (Subramani et al., 2021).

## 7 MULTI-GPU TRAINING

This subsection will look at another angle to train deep learning with DP faster: increasing the computational resources enough to decrease the training time. This is relevant when training cost or resource constraints are less important than the time to train a new model.

We utilize V100 GPUs on HPC nodes that have 4 GPUs per node. The other experimental setting is identical to the one in Section 4. Results for utilizing up to 24 A100 GPUs can be found in Figure A.5 in the Appendix. We focus on comparing the scaling behaviour between the non-private baseline that uses PyTorch and the DP-SGD implementation using Opacus. Both frameworks provide mature tooling for distributed training.

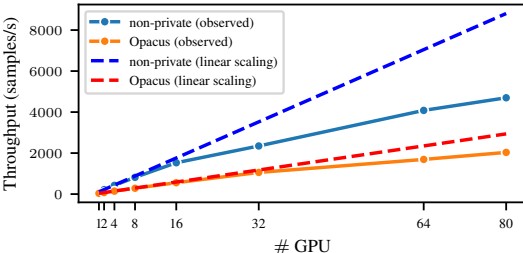

Figure 7: Comparison between the throughput by scaling the number of GPUs with more nodes for the non-private and Opacus training with the ViT base model on V100 GPUs. The dashed line is the ideal growth if it were linear.

Figure 7 shows the throughput increase as a function of number of GPUs. The throughput does not grow linearly and changes from the ideal linear scaling after using more than one node (i.e. when using more than 4 GPUs). The communication inside the node is fast, but the communication between nodes will always be slower. The bottleneck is the bandwidth, and it prevents the model from scaling linearly. Notably, it affects the non-private training baseline much more, while the private scales close to optimal up to 32 GPUs. For the 80 GPUs, the private training achieves $69.2\%$ of the ideal throughput, and the non-private training only achieves $53.3\%$. Private training scales better because it is slower and only sometimes saturates the network with updates.

If we use Amdalh's law to compare the parallelism percentage for each case, we can see that in the private case, we achieve a $99.5\%$ parallelism compared to a $98.9\%$ in the non-private case (See Figure A.6 in the Appendix).

## 8 CONCLUSION

Table 5: A summary of the lessons learnt. The relative throughput/max physical batch size is in comparison to PyTorch non-DP (higher is better) on NVIDIA A100. For each optimization method and each model size, we divide it with the non-private counterpart.

| Method | Relative to non-DP (PyTorch) | | Supports all layers | Compilation | | Privacy Concerns | Section |
| --- | --- | --- | --- | --- | --- | --- | --- |
| | Throughput (↑) | Max Physical Batch Size (↑) | | Initial | Re- | | |
| Opacus | 0.31-0.39 | 0.08-0.24 | ✓ | - | - | - | Section 4 |
| Efficient Gradient Clipping | 0.49-0.54 | 0.88-0.95 | ✗ | - | - | - | Section 5.1 |
| Native JAX | 0.39-0.59 | 0.23-0.43 | ✓ | ✓ | ✓ | - | Section 6 |
| Masked DP-SGD (ours) | 0.51-0.69 | 0.11-0.23 | ✓ | ✓ | ✗ | - | Section 6 |
| Masked DP-SGD + TF32 | 0.79-1.33 | 0.11-0.23 | ✓ | ✓ | ✗ | ? | Section 6 |
| Low Precision (Opacus+TF32) | 0.54-0.84 | 0.08-0.24 | ✓ | - | - | ? | Section 5.2 |

We summarize the lessons learnt in Table 5. While DP-SGD is significantly more costly than non-private training, we identified feasible speed-ups that are often easy to apply but have some drawbacks. These are: (i) More efficient implementations of DP-SGD which additionally decrease the memory footprint (allowing for training larger models). However, these implementations are not as mature as Opacus and do not support all neural network layers (yet). (ii) JAX which processes samples faster than PyTorch but looses the advantage through frequent re-compilations when utilizing proper Poisson sampling. Moreover, JAX lacks a comprehensive DP-SGD implementation as PyTorch and exhibits a greater variability in execution times. (iii) We present an efficient implementation DP-SGD with JAX that correctly uses Poisson sampling while using physical batches of the same length, also complying with JAX efficient optimizations. (iv) Lower Precision using TF32 which increases throughput but the implications on the theoretical guarantees of DP-SGD need to be explored in future work. Finally, we found that distributed computing using DP-SGD scales better than non-private training and allows for fast training of models.

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

## A  TRAINING DETAILS

### A.1  MODELS

- Vision Transformer (ViT) (Dosovitskiy et al., 2021). Taken from `https://huggingface.co/timm/vit_base_patch16_224.orig_in21k`
- Big Transfer ResNet (Kolesnikov et al., 2020). Taken from `https://github.com/google-research/big_transfer`

### A.2  HYPERPARAMETERS

We use the hyperparameters obtained on request from Tobaben et al. (2023). The hyperparameters for both models are in Table A1. Even though model utility is not the main objective in this work, in the non-private case, the learning rate is suboptimal. By changing it to 0.00027 we see an accuracy improvement, therefore the one we are using.

Table A1: Hyperparameters used for each model architecture.

| MODEL | TRAINABLE PARAMETERS | EPSILON | DELTA | LEARNING RATE | MAX GRAD NORM |
|-------|----------------------|---------|-------|---------------|---------------|
| ViT | ALL | 8 | $2.04e^{-5}$ | 0.00031 | 4.63 |
| RESNET | ALL | 8 | $2.04e^{-5}$ | 0.00098 | 6.53 |

### A.3  GRAD SAMPLE MODES IN OPACUS

Opacus supports multiple different gradient sampling methods as indicated in the documentation[1]. In our original experiments we used the grad_sample mode *hooks* that is the default. This will use

---

[1] `https://github.com/pytorch/opacus/tree/61ae0ea4fb37a835e93040b5de19e8dfcd465a07/opacus/grad_sample`

custom opacus modules when they are defined for that layer and functorch as a fallback. Based on the feedback by a reviewer we tried out different methods listed in the documentation for both ResNet and ViT models:

- $functorch$: We forced opacus to use functorch but did not observe any significant speed differences to using $hooks$. This is in line with the opacus documentation which writes that the speed is $0 - 50\%$ slower than $hooks$.
- $ExpandedWeigths$: We tried this approach but ran into runtime errors. Interestingly, when looking through the issues others have reported issues[2][3] but it seems to be more a PyTorch problem and has not been addressed for years. According to the documentation $ExpandedWeights$ is still in beta status.
- $GhostClipping$: Note that this method only works for ViT as described in Section 5.1. We did not manage to decrease the loss with this implementation due to the implementation in opacus being unstable but think that the speedups should be similar as observed in our experiments in Section 5.1 as the underlying algorithm is the same.

### A.4 POISSON SUBSAMPLING JAX ALGORITHM

We present our DP-SGD implementation in JAX that uses the correct Poisson subsampling and therefore we can account for its privacy. The main problem with implementing DP-SGD with JAX is the batches of variable size. In order to address this issue, we compute always full physical batches and mask out gradients so that the total number of used gradients is equal the sampled logical batch sizes. This means that we always compute a little more gradients that required due to sampling. This prevents the recompiling.

---

**Algorithm 2** Virtual Batching DP-SGD JAX

---

**Input:** Training data points $\{x_1, \ldots, x_N\}$, loss function $\mathcal{L}(\theta) = \frac{1}{N}\sum_i \mathcal{L}(\theta, x_i)$
**Parameters:** Parameters: learning rate $\eta_t$, noise scale $\sigma$, gradient norm bound $C$, number of steps $T$, expected logical batch size $L$, physical batch size $p$.
**Start**
**for** $t \in [T]$ **do**
  $tl \sim \text{Bernoulli}(\frac{L}{N})$ {Sample the true batch size}
  Find minimum $k \in \mathbb{N}$ such that $p \cdot k \geq tl$ {Check how many full physical batches are required}
  $m \leftarrow k \cdot p$
  $B \leftarrow \{x_{j_1}, \ldots, x_{j_m}\}$
  $P \leftarrow \{B_1, \ldots, B_k\}$ {Divide the maximum logical batch $B$ into physical batches of size $p$}.
  $M \leftarrow \{1_0, 1_1, \ldots, 1_{tl-1}, 0, 0, \ldots, 0_{m-tl+1}\}$ {Create masks so that $\sum_i^m M_i = tl$}
  $\theta_{acc} \leftarrow \mathbf{0}$
  **for** $s \in [P]$ **do**
    **for** $i \in s$ **do**
      $\mathbf{g}_t(x_i) \leftarrow \nabla_{\theta_t}\mathcal{L}(\theta_t, x_i)$ {**Compute gradient**}
      $\overline{\mathbf{g}}_t(x_i) \leftarrow M_{i+(s-1)*p} \cdot \mathbf{g}_t / \mathbf{max}(1, \frac{\|\mathbf{g}_t(x_i)\|_2}{C})$ {**Clip gradient and mask**}
    **end for**
    $\theta_{acc} \leftarrow \theta_{acc} + \sum_i \overline{\mathbf{g}}_t(x_i)$ {**Accumulate gradient**}
  **end for**
  $\widetilde{\mathbf{g}}_t \leftarrow \frac{1}{|L|}(\theta_{acc} + \mathcal{N}(0, \sigma^2 C^2 \mathbf{I}))$ {**Add noise**}
  $\theta_{t+1} \leftarrow \theta_t - \eta_t \widetilde{\mathbf{g}}_t$ {**Step**}
**end for**
**Return** Learned parameters $\theta_T$ and the privacy cost from a privacy accountant.

---

## B ADDITIONAL RESULTS

This section provides additional figures that supplement the findings in the main text.

---

[2] https://github.com/pytorch/opacus/issues/464
[3] https://github.com/pytorch/opacus/issues/584

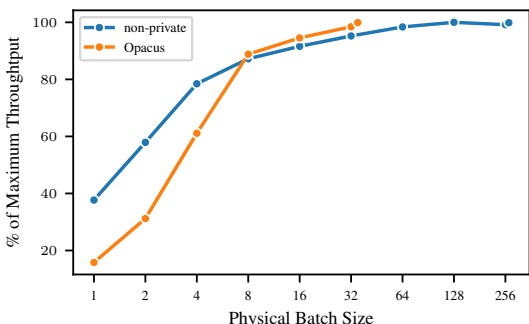

Figure A.1: The relative difference with the throughput at the maximum batch size for the ViT base model on A100.

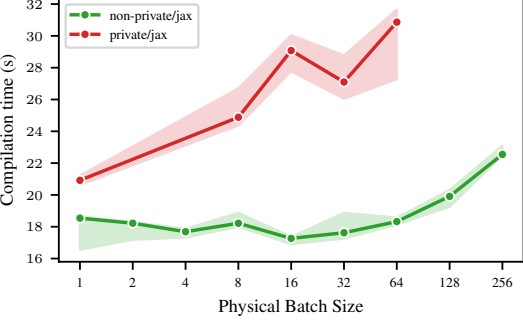

Figure A.2: Compilation time in seconds as a function of the physical batch size for JAX naive experiments for the ViT Base model on A100. The estimator is the median and the error bars are the 95% confidence interval using bootstrapping.

Table A2: Mean accuracy for CIFAR-100 test set for each clipping mode for the ViT models on A100 after training for two epochs. All use the ViT hyperparameters from Table A1. While this work does not focus on the model's utility, having their results still allows us to compare them. The use of TF32 as a lower precision mode does not affect the model's utility.

| CLIPPING MODE | TEST ACCURACY |
|---|---|
| OPACUS | 0.8223 |
| OPACUS/TF32 | 0.8225 |
| JAX NAIVE | 0.8146 |
| MASKED DP-SGD | 0.8224 |
| PV-GHOST | 0.822 |
| BK-GHOST | 0.822 |

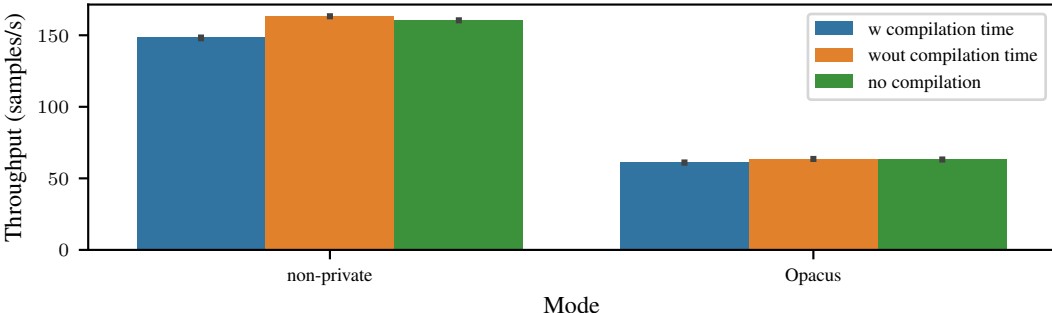

Figure A.3: Torch compilation experiments on A100, using the maximum physical batch size for each mode and ViT Base. PyTorch 2 enables compiling the model to (potentially) gain further speed-ups. We tried PyTorch 2 compilation to make a fair comparison with the JAX compilation but did not observe any benefits from it. We found that when trying to compile PyTorch, it first tries to compile but then falls back to NVIDIA kernels and optimizations. In the end, it does not compile, and the throughput is the same. If we take into account the first iteration (w compilation time), it is worse because of the time PyTorch spends trying to compile before falling back to NVIDIA kernels and optimizations. Disregarding the time where PyTorch tries to compile (wout compilation time), leads to nearly the same throughput as the version that does not attempt using PyTorch 2 compiling in the first place.

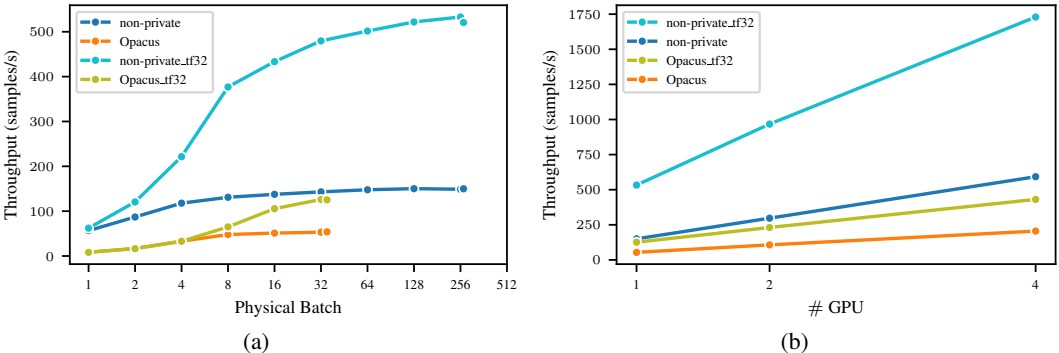

Figure A.4: Combining distributed training with the use of lower precision TF32 for the ViT base model on A100. (a) Throughput for one GPU; (b) Throughput for multiple GPUs.

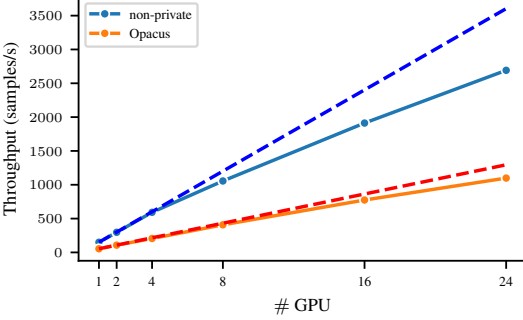

Figure A.5: Comparison between the throughput by scaling the number of GPUs with more nodes for the non-private and Opacus training with the ViT base model on A100 GPUs. The dashed line is the ideal growth if it were linear.

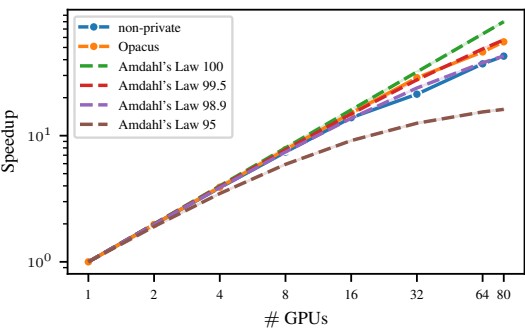

Figure A.6: Comparison between the throughput in our experiments and the theoretical Amdahl's Law. Both axis are in log scale. In the distributed setting, private training achieves a 99.5 % of parallel processing, with a 50 times speed up than single processing.

## C   FURTHER DISCUSSION OF TF32 SPEEDUPS

The speedup observed in Figure 5 peaks at the "base" model. We believe that the reasons are the following: Speed-ups resulting from TF32 can significantly vary on per case basis as "all storage in memory and other operations remain completely in FP32, only convolutions and matrix-multiplications convert their inputs to TF32 right before multiplication." (Stosic & Micikevicius, 2021). Until now, TF32 precision benchmarks have been limited to non-DP applications which was one of the reasons we wanted to discuss our observations in DP context. It appears the effectiveness of TF32 arithmetic peaks at "base" configuration. This due to a mix of reasons which are difficult to quantify exactly. Firstly, it is likely that matrix multiplication kernel dominance peaks at this configuration i.e. we have the most parameters whilst the batch size dimension also remains sufficiently large. With large and huge model variants the parameter count still increases but at the cost having very small batch dimension of 10 and 3, respectively. Secondly, we observe similar trend in Figure 2(a) where the discrepancy between dp and non-dp grows as model size gets bigger. This suggests that the dominance of DP operations also grows with the model size. None of the DP-operations are cast as matrix-multiplications and hence won't benefit from TF32.

## D   EXTRA COMPUTATIONAL COST OF THE `MASKED DP-SGD`

For the `masked dp-sgd`, we first sample the minibatch using Poisson subsampling and to allow JAX compilation, we round this number to the closest larger integer divisable by the physical batch size. Hence, for any samples batch size $X$, the difference between $X$ and the upscaled batch size will be in $\{0, \ldots p-1\}$ for a physical batch size $p$. Denoting the excess batch size with $\Delta_p(X)$ and the upscaled batch size with $X_+$, we can write

$$\mathbb{E}[X_+] = \mathbb{E}[X + \Delta_p(X)]. \tag{A1}$$

Now, we can form a simple upper bound for the expected value of the upscaled batch size as

$$\mathbb{E}[X_+] \leq \mathbb{E}[X] + (p-1). \tag{A2}$$

When working large number of samples and non-negligible sampling probabilities, the excess cost due to upscaling the batch size will be modest for feasible physical batch sizes. For example, in our experiments the expected batch size of the Poisson subsampling was $25\,000$, whereas the physical batch sizes extended up to $64$.

A recent work by Chua et al. (2024b) proposed an alternative implementation for JAX compilable implementation of Poisson subsampled DP-SGD. In their approach the batch sizes are sampled from a truncated Binomial distribution. This affects the privacy guarantees of the models, and therefore they need to compensate the truncated sampling by increasing the noise std. for DP-SGD. They suggest an approach for computing the truncation bound $B$ as

$$\Psi(n, b, B) \cdot T \cdot (1 + e^\epsilon) \leq \tau\delta \tag{A3}$$

where $\Psi(n, b, B)$ denotes the survival function $(1 - \texttt{cdf})$ of $\mathrm{Binom}(n, b/n)$ at $B$ and $T$ are the number of steps. The parameter $\tau$ effectively scales the size of the tails and is used to calibrate the noise std by selecting $\sigma$ such that the hockey-stick divergence between the Poisson subsampled Gaussian mechanisms is bound by $(1-\tau)\delta$. Chua et al. (2024b) choose $\tau = 10^{-5}$, which keeps the noise std. increase very small.

In the implementation of Chua et al. (2024b), the gradients are computed for $B$ randomly selected samples, after which the final samples are chosen according to the batch size sampled from the truncated Binomial. Hence the computational excess over regular Poisson subsampling becomes $B - b$. For example, in our setting where $\epsilon = 8$, $\delta = 10^{-5}$, $n = 50\,000$, $b/n = 1/2$ and $T = 4$, the $B - b = 858$, which is significantly larger than the $p - 1$ excess of our method for obtainable physical batch sizes ($p \leq 64$).

