# OpenReview forum: "Towards Efficient and Scalable Implementation of Differentially Private Deep Learning"
_ICLR.cc/2025/Conference — Submitted to ICLR 2025_

### Official Review · Reviewer_ZyeT · 2024-10-28

**Soundness:** 3
**Presentation:** 1
**Contribution:** 2
**Rating:** 3
**Confidence:** 4

**Summary:**

This paper investigates the computational cost and efficiency of DP-SGD training, emphasizing that many current implementations forego proper Poisson subsampling, which could result in weaker privacy guarantees than intended. The authors propose an alternative, more efficient DP-SGD implementation in JAX with Poisson subsampling, achieving throughput only 1.2 times lower than standard PyTorch-based SGD. Besides this, the author also attempt to reveal the reason behind the high training cost and the scaling behavior using a larger clusters of hardware resources.

**Strengths:**

1. The authors re-implement DP-SGD in a more efficient way and the source code is shared in the supplementary materials.
2. The authors discuss various types of gradient clippings.
3. The authors investigated the reason of high training cost by the NVIDIA Nsight System.

**Weaknesses:**

1. The paper is written unprofessionally, unstructured and very hard to follow.
2. The motivation is unclear. Based on the abstract and introduction, the paper starts from the issue of weaken privacy guarantees in current improper implementations of Poisson subsampling. However, the rest of the paper discuss the computation cost of DP-SGD, which is unrelated. After reading the whole manuscript, the reviewer still does not know how, whether the authors address this problem or not.
3. The background of implementations of DP-SGD is poor written. In Sec. 2.1, the paper quickly jumps into Virtual batching and discuss some detailed batch sizes (25000, 300, etc), which seems to be more appropriate in Sec. 3. How are paragraphs of Sec. 2.1 relate to each others? Are they parallel or progressive? The reviewer find them unstructured. Virtual Batching, Poisson subsampling are high-level algorithm idea, and Opacus is a software package. The reviewer does not think they are appropriate to be listed together. The rest of the paper has the same issue, where the authors list a bunch of unstructured items in a section, which are very hard to follow.
4. The contributions are poorly summarized in the introduction section. The reviewer understands that the authors have investigated on various aspects of the cost, but still finds unclear what the main points are here.
5. The novelty remains a major drawback of this paper. First, the author acknowledged that their current implementation is incomplete and does not support all network types yet. Second, using JAX or lower precision for speed up training is very common and too naive, even in other generic non-private training. Third, the paper raises some new questions from their empirical findings, while the authors do not address them. Based on these 3 points, the reviewer thinks the paper is incomplete.

**Questions:**

1. The authors need to present their proposed implementation and explain how, whether it insures correct privacy guarantee of Poisson subsampling,
2. How algorithms are optimized in Sec. 5.2 to decrease the cost? From the writing, it seems that the way to "optimize" is just using a better GPU (A100 instead of V100).

---

> ### Author Response · Authors · 2024-11-21
> **Answer to reviewer ZyeT**
>
> > paper starts from the issue of weaken privacy guarantees in current improper implementations of Poisson subsampling. However, the rest of the paper discuss the computation cost of DP-SGD, which is unrelated.
>
> Very recent work [3,4] (ICML 2024/NeurIPS 2024) has shown that basing the accounting on Poisson subsampling while implementing a different sampling strategy can lead to a significant difference in the actual privacy loss compared to the accounted one. Especially noteworthy is Figure 3 (middle) of [4] which illustrates the required $noisemultiplier$ for the different sampling mechanisms with the Poisson subsampling $noisemultiplier$ being constantly lower than the others. Furthermore, [4] shows that Poisson subsampling is crucial in achieving the optimal utility under DP (Figure 3, left) while other sampling mechanisms can be beneficial in the non-DP setting (Figure 3, right). Another very recent paper [5] (preprint available on 15 November) also showcases this problem in their Figure 1. We already cited the ICML paper [3] but we added a citation to the NeurIPS paper [4] and the arXiv pre-print [5] in the revision.
>
> Implementing Poisson subsampling is not trivial as Poisson subsampling results in varying (logical) batch sizes at every step of DP-SGD. This is especially challenging for the JAX implementations as changing tensor sizes (can) trigger recompilations as also mentioned by [4]. We believe that this is the reason why prior JAX implementations do not correctly implement the Poisson subsampling but instead use other sampling methods while using the privacy accounting for Poisson subsampling. We would like to clarify that none of the original versions of the compared implementations apart from Opacus correctly implements Poisson subsampling and that we add the Poisson subsampling to these to allow for a fair comparison of throughput and maximum possible physical batch size. Furthermore, we propose the masked DP-SGD algorithm as an alternative to the naive implementation of Poisson subsampling in JAX that is prone to recompilations.
> > In Sec. 2.1, the paper quickly jumps into Virtual batching and discuss some detailed batch sizes (25000, 300, etc), which seems to be more appropriate in Sec. 3. How are paragraphs of Sec. 2.1 relate to each others?
>
> Thanks for the feedback. We restructured the Sections 2 and 3 to improve the readability of them and hope that these changes address your criticism. The mentioned batch sizes are for illustrative purposes and we added a “For example” to the sentence to make that clearer.
>
> >  First, the author acknowledged that their current implementation is incomplete and does not support all network types yet.
>
> We believe there is a misunderstanding as some of the efficient algorithms like GhostClipping do not support all layers, but this is an algorithmic limitation not something that can be trivially fixed. We do not propose these algorithms but benchmark all of them under the requirement of using Poisson subsampling. We would like to highlight that our own contribution (masked DP-SGD implementation in JAX) should support all layers and we are not aware of any limitations regarding that.
>
> > Second, using JAX or lower precision for speed up training is very common and too naive, even in other generic non-private training
>
> The implementation using Poisson subsampling is not straightforward as the naive implementation in JAX suffers from recompilations (see also [4] and that can be overcome using our proposed method masked DP-SGD).
>
> > The authors need to present their proposed implementation and explain how, whether it insures correct privacy guarantee of Poisson subsampling,
>
> Based on your feedback and the feedback of other reviewers we improved the writing of Section 2 and 3 and added a short paragraph about our proposed method. We agree with you that this was not written clearly enough in the first version of the paper.
>
> > How algorithms are optimized in Sec. 5.2 to decrease the cost? From the writing, it seems that the way to "optimize" is just using a better GPU (A100 instead of V100).
>
> Thanks to the reviewer for the comments. We understand that there is some confusion about it. We changed the Section 5.1 title from Optimized gradient clipping algorithms, to Efficient gradient clipping algorithms, to highlight that the algorithms (Ghost Clipping, Mix Ghost Clipping, Opt Mix) by themselves are an efficient way of calculating the gradients and present an alternative to the Opacus implementation, but that we are not optimizing them.
>
> [3] Chua, L., Ghazi, B., Kamath, P., Kumar, R., Manurangsi, P., Sinha, A., & Zhang, C. How Private are DP-SGD Implementations?. ICML 2024
>
> [4] Chua, L., Ghazi, B., Kamath, P., Kumar, R., Manurangsi, P., Sinha, A., & Zhang, C. Scalable DP-SGD: Shuffling vs. Poisson Subsampling. NeurIPS 2024
>
> [5] Annamalai, MSMS, Balle, B, De Cristofaro, E, Hayes, J. To Shuffle or not to Shuffle: Auditing DP-SGD with Shuffling. arXiv:2411.10614

---

> > ### Comment · Reviewer_ZyeT · 2024-11-25
> >
> > Thank you for your response. I now have a clearer understanding of the motivation behind the paper after your revision, but your changes should be highlighted with different colors, which would make them more noticeable.
> >
> > I believed that the presentation is still a big problem. For example, your proposed method (Masked DP-SGD) is not introduced until Section 3 (Experiment), which limits the readability of the paper. It would be better to mention and briefly describe it earlier in the introduction. Table 1 should include citations for the baseline methods to provide proper attribution. In Table 3, I assume the baseline *BK MIX OPT* refers to *Book Keeping + Mix Optimization*. Have you clearly explained all the abbreviations (*BK*, *PV*, etc.) used in the paper? If not, adding these explanations would enhance clarity. I understand that it might not be easy to organize existing tools, but there is still significant room for improvement in the presentation.
> >
> > The contributions of Masked DP-SGD is limited, because it just simply adds a few ghost samples and later drops their gradients, so that it can accommodate a fixed batch size for compilation purposes. From the perspective of software development, it might be challenging, but it brings limited insights regarding the understanding of privacy guarantees of current DP implementations.
> >
> > As a benchmark paper, this paper is not comprehensive enough. The design of the entire experiment, as well as all the compared methods, can be explained in greater detail in the appendix. Particularly, the paper does not benchmark the performance of all methods, including the proposed method. It is necessary to provide it at least for your own method, otherwise we could not know the correctness of your implementation. I have noticed two other reviewers (gRw1, srbv) raised the same concern, but the author did not answer (unless I overlooked it).
> >
> > 1. You said that *none of the implementations but Opacus correctly implements Poisson subsampling and that we add the Poisson subsampling to these to allow for a fair comparison*. Do you mean that all the methods your benchmark in Figure 4, Table 3, and Figure 7 are implemented with Masked DP-SGD except Opacus?
> >
> > 2. What does *o-flat* refer to in Figure 3 and 7?
> >
> > 3. Does Masked DP-SGD require more GPU memory and training time than standard DP-SGD?
> >
> > 4. Is the lower precision benchmarked using Opacus? Did you benchmark lower precision with Mask DP-SGD? If not, why not and what is the purpose of Section 5.2?
> >
> > 5. It is better to compare with other SOTA methods you mentioned, e.g. truncated Poisson subsampling in [1].
> >
> > [1] Chua, L., Ghazi, B., Kamath, P., Kumar, R., Manurangsi, P., Sinha, A., & Zhang, C. Scalable DP-SGD: Shuffling vs. Poisson Subsampling. NeurIPS 2024

---

> > > ### Author Response · Authors · 2024-11-26
> > > **Answer to Reviewer ZyeT (Part 1/2)**
> > >
> > > > your changes should be highlighted with different colors, which would make them more noticeable.
> > >
> > >
> > > Thanks for the suggestion, we highlighted all changes in the text in comparison to the original submission in blue.
> > >
> > >
> > > > Table 1 should include citations for the baseline methods to provide proper attribution
> > >
> > >
> > > Thanks for the suggestion, we added the citations to Table 1.
> > >
> > >
> > > > In Table 3, I assume the baseline BK MIX OPT refers to Book Keeping + Mix Optimization.
> > >
> > >
> > > Yes! As we explain in Section 2.2 DP-SGD gradient clipping optimizations, the Book Keeping algorithm can also be combined with the mix optimization. Thanks for the comment, because now with the citations on the table, it is clearer that all the methods implemented by the FastDP [10] paper use Book Keeping.
> > >
> > >
> > > [10] Zhiqi Bu, Yu-Xiang Wang, Sheng Zha, and George Karypis. Differentially Private Optimization on Large Model at Small Cost. ICML 2023
> > >
> > >
> > >
> > >
> > > > Have you clearly explained all the abbreviations (BK, PV, etc.) used in the paper?
> > >
> > >
> > > Thanks for the comment. We took into account Table 1 and Section 3. There, we add the abbreviations of BK and PV, which depend on the library.
> > >
> > >
> > > > The contributions of Masked DP-SGD is limited, because it just simply adds a few ghost samples and later drops their gradients, so that it can accommodate a fixed batch size for compilation purposes.
> > >
> > >
> > > We would like to highlight again that our proposed method (Masked DP-SGD) allows for combining Poisson subsampling (required to get optimal privacy/utility trade-offs as discussed by referenced work) with frameworks that rely on compiling (like JAX). We believe it is a valuable contribution as in the settings that we consider it is more efficient than concurrent work published at NeurIPS 2024. (See your question below).
> > >
> > > > You said that none of the implementations but Opacus correctly implements Poisson subsampling and that we add the Poisson subsampling to these to allow for a fair comparison. Do you mean that all the methods your benchmark in Figure 4, Table 3, and Figure 7 are implemented with Masked DP-SGD except Opacus?
> > >
> > >
> > > Thanks to the reviewer for the question, and we addressed it in Section 3 Experiment Overview in the paragraph “Implementation of Poisson subsampling”. There we highlight that the other frameworks that implement Ghost Clipping and Book Keeping do not support virtual batching out-of-the-box and in their code they do not use proper poisson subsampling. Therefore we implement it for them, in a similar way as Opacus does.
> > >
> > > We do not claim that the only correct implementation of Poisson subsampling is the one we made for our Masked DP-SGD and also it is important to differentiate between the sampling method and the clipping method. Our Masked DP-SGD implements Poisson subsampling specifically for the JAX framework, but still uses the original per-example clipping. We do not implement Masked DP-SGD, which masks the gradients, for PyTorch, since we want to overcome the disadvantages that Poisson subsampling presents in a framework with compilation like JAX.
> > >
> > > > What does o-flat refer to in Figure 3 and 7?
> > >
> > >
> > > Thanks for pointing out this mistake, this is a typo. We changed “o-flat” to “Opacus” in Figures 3 and 7. (“O-Flat” is used in the source code to denote flat clipping with Opacus - the default method used in Opacus that we benchmark).
> > >
> > >
> > > > Does Masked DP-SGD require more GPU memory and training time than standard DP-SGD?
> > >
> > > To answer this question we can refer to Figure 6. In it, our method, Masked DP-SGD reaches a higher physical batch size than the baseline Opacus method. It almost doubles it, Opacus higher physical batch size is 35, ours is 64. So it is more memory efficient, but not as much as other clipping methods. Where we see the biggest impact is in the training time. We are using the throughput as samples per second. Our implementation is the one that is closest to the non-private baseline, much better than the private Opacus version. It is 2.2 times faster than Opacus.
> > >
> > > > Is the lower precision benchmarked using Opacus? Did you benchmark lower precision with Mask DP-SGD? If not, why not and what is the purpose of Section 5.2?
> > >
> > >
> > > Thanks for pointing this out. To answer the first and last questions, yes, the original use of lower precision is to make the slower method, Opacus, faster. To see the gains of using a lower precision mode, which in theory should be faster, to overcome the non-trivial cost of DP-SGD. That is the purpose of Section 5.2, highlighting the benefits of using lower precision, which makes per-example clipping by Opacus 2 times faster.

---

> > > > ### Author Response · Authors · 2024-11-26
> > > > **Answer to Reviewer ZyeT (Part 2/2)**
> > > >
> > > > Here we continue the replies.
> > > >
> > > > > Is the lower precision benchmarked using Opacus? Did you benchmark lower precision with Mask DP-SGD? If not, why not and what is the purpose of Section 5.2?
> > > >
> > > >
> > > > Thanks for pointing this out. To answer the first and last questions, yes, the original use of lower precision is to make the slower method, Opacus, faster. To see the gains of using a lower precision mode, which in theory should be faster, to overcome the non-trivial cost of DP-SGD. That is the purpose of Section 5.2, highlighting the benefits of using lower precision, which makes per-example clipping by Opacus 2 times faster.
> > > >
> > > >
> > > > > Did you benchmark lower precision with Mask DP-SGD?
> > > >
> > > >
> > > > Thanks for your question. Based on your question we revised the paper to include both TF32 and FP32 with JAX to Figures 1 and 6 and Section 6. We believe that these results enrich the discussion for the use of lower precision methods.
> > > >
> > > >
> > > > We previously did not focus on the precision for JAX but now thanks to your pointer found out that (counterintuitively) JAX uses TF32 without warning by default with A100 [11] while we would have expected that FP32 is the default as in other frameworks. “On GPU: uses tensorfloat32 if available (e.g. on A100 and H100 GPUs), otherwise standard float32 (e.g. on V100 GPUs)”. Our revised manuscript now makes clear which precision mode is used in the comparison.
> > > >
> > > > [11] https://jax.readthedocs.io/en/latest/jax.lax.html#jax.lax.Precision
> > > >
> > > > > It is better to compare with other SOTA methods you mentioned, e.g. truncated Poisson subsampling in [1].
> > > >
> > > > Thanks for the comment. [1] is concurrent work (see explanation below) and we are unaware of any other works solving the JAX recompiling problem with Poisson subsampling. We added a pen and paper comparison to [1] to Appendix D which showcases that our approach is more efficient for the logical and physical batch sizes considered in our setting.
> > > >
> > > > **Current work:** We would like to highlight again that [1] got available on arXiv on 06. November 2024 which is the same time that the NeurIPS version got released to the public. We view [1] as concurrent work as it got released after the submission deadline (01 October 2024), see also [ICLR 2025 FAQs].
> > > >
> > > > **Pen and paper comparison**: Unfortunately, [1] do not release any source code, so we are unable to make full runtime comparisons in the short time left in the discussion period but instead decided to make a pen and paper comparison that can be found in Appendix D. In short we show that for example, in our setting where $\epsilon=8$, $\delta=1e-5$, $n = 50 000$, subsampling ratio of 0.5, $T = 4$, and the physical batch size is 64, the excess of [1] is 858, which is significantly larger than the p − 1 excess of our method for obtainable
> > > > physical batch sizes (p ≤ 64).
> > > >
> > > > [ICLR 2025 FAQs] https://iclr.cc/Conferences/2025/FAQ

---

> > > > > ### Comment · Reviewer_ZyeT · 2024-11-27
> > > > >
> > > > > Thank you for the detailed explanation. I believe the paper is now clearer in terms of the connections between sections and the combinations of different implementation methods. I appreciate the inclusion of a comparison to *truncated Poisson subsampling*, even though it is a concurrent work and does not require direct comparison. I also appreciate the changes made regarding lower precision and the separation of Figures 6(a) and 6(b). From Figure 6(b), it can be observed that, although Masked DP-SGD achieves similar throughput to Opacus under TF32, it allows training with a larger batch size.
> > > > >
> > > > > I keep my score for the following reasons,
> > > > >
> > > > > 1. I remain somewhat unconvinced about the level of impact of the proposed method, as it provides limited insight into current DP implementations with Poisson subsampling.
> > > > >
> > > > > 2. As a benchmark paper it is not comprehensive enough. Three reviewer (gRw1, srbv, me) repeatedly asked the same question (reporting the performance), but the author did not provide an answer. This is necessary for verifying the correctness of the proposed method. Other details about the experiment are also expected.
> > > > >
> > > > > 3. The presentation remains as a weakness for the paper. I understand the authors have done multiple experiments. However, I find it difficult to connect them to the main point of the paper. For example, Section 7 evaluate Opacus only on multi-gpu, but Masked DP-SGD is not included. I have no idea how this relates to the proposed method. All baseline methods to compare can be listed properly somewhere in the middle of the paper, and then stick to the same abbreviation for the rest of the paper. I strongly recommend the author to reconsider the entire paper,  focusing on the main points of your idea. (overall structures, connections between sections, minor issues, etc).

---

> > > > > > ### Author Response · Authors · 2024-11-27
> > > > > >
> > > > > > Thanks for your answer. We appreciate your time and feedback.
> > > > > >
> > > > > > >  Three reviewer (gRw1, srbv, me) repeatedly asked the same question (reporting the performance), but the author did not provide an answer. This is necessary for verifying the correctness of the proposed method. Other details about the experiment are also expected.
> > > > > >
> > > > > > Could you please elaborate on this point. What do you mean by performance? We report throughout and maximum physical batch size for all methods.
> > > > > >
> > > > > > We report utility at privacy numbers in the Appendix but this is mostly to confirm that the implementations are correct. In principle all methods should achieve the same utility numbers when using the same precision. What other details do you think are missing?

---

> > > > > > ### Author Response · Authors · 2024-11-28
> > > > > >
> > > > > > > Three reviewer (gRw1, srbv, me) repeatedly asked the same question (reporting the performance), but the author did not provide an answer. This is necessary for verifying the correctness of the proposed method.
> > > > > >
> > > > > > Thanks for pointing that out. Please see the general response "Regarding the utility/privacy trade-off (Table A2)" above.

---

> > > > > > ### Author Response · Authors · 2024-11-28
> > > > > > **Regarding the distributed JAX experiments**
> > > > > >
> > > > > > >  For example, Section 7 evaluate Opacus only on multi-gpu, but Masked DP-SGD is not included.
> > > > > >
> > > > > > We would like to point our that Section 7 compares the scaling of non-private PyTorch and private PyTorch (Opacus). As pointed out in the beginning of Section 7 we focus on these two as they provide mature tooling for distributed training. We agree with the reviewer that adding a comparison to the scaling of masked DP-SGD would be insightful but would like to note that we are unable to fully implement distributed masked DP-SGD (in JAX) and benchmark it (using up to 80 GPUs) during the relatively short time of the rebuttal.
> > > > > >
> > > > > > We believe that the general findings regarding the scaling behaviour due to different bottlenecks is interesting and think that potential findings would transfer to our implementation as the general properties in regards in ratio of time to communicate over the network vs. local computation should be similar as with Opacus although due to the the higher throughput of masked DP-SGD we would expect a poorer scaling behaviour than Opacus. Note that the total throughput would still be expected to be higher for masked DP-SGD in comparison to Opacus but the relative benefit of using more GPUs might be smaller.
> > > > > >
> > > > > > We are unable to revise the paper further but we would be happy to add this insightful and important discussion to the final version of the paper if the reviewer agrees that this would be beneficial.

---

> > > > > > > ### Comment · Reviewer_ZyeT · 2024-12-02
> > > > > > >
> > > > > > > The reviewer is aware that Table A2 presents the test accuracy after training for four steps. The reviewer is also aware that the paper does not benchmark the proposed method, masked DP-SGD, in Section 7 and reports only the baseline method, Opacus.

---

### Official Review · Reviewer_L1Bo · 2024-11-01

**Soundness:** 3
**Presentation:** 4
**Contribution:** 3
**Rating:** 8
**Confidence:** 3

**Summary:**

The authors present an extensive empirical study of how efficient current proper (i.e., with poisson sampling) DP implementations are, and then provide an algorithm that leverages JAX better to improve the throughput of DP training. I believe the empirical study and modifications are interesting and impactful for the field; scalable DP training is an important technical challenge for private learning, and they provide detailed understanding for current computational issues with DP implementations and ways of bypassing some of these. Furthermore, I found the paper to be generally well written.

**Strengths:**

1) Extensive empirical study
2) Proposed algorithm improves previous approaches
3) Well-written: I expect this will help with future work on the topic

**Weaknesses:**

1) One could argue the algorithmic contributions of the paper are limited, being just modifications to improve sampling. However, I believe it required a detailed study to realize that Jax could help if one tuned away the reocurring compilation times, paying a modest increase in gradient evaluations. In the questions I also ask the authors to surface more details on the algorithm they propose to the main text, which may help convey the contributions more.

**Questions:**

1) In the caption for figure 2, is there a typo when saying lower throughput is better?
2) In lines 399 to 400, could you elaborate on why larger models are too expensive (to have benefit from lower precision)?
3) In the JAX paragraph of section 2.2, could you elaborate on how masking achieves poisson sampling? I found the algorithm in the appendix, and I think describing it briefly and pointing to the algorithm in the appendix would help with clarity on this contribution: e.g.,  something along the lines "Specifically, we first sample a batch size from the distribution of batch sizes, and with this sample a number of random physical batches to have more samples: one can then mask out some datapoints to have effectively sampled with poisson."

---

> ### Author Response · Authors · 2024-11-21
> **Answer to reviewer L1Bo**
>
> > In the caption for figure 2, is there a typo when saying lower throughput is better?
>
>
> Thanks for pointing out this unclarity! We clarified this in the paper and renamed the y-axis-label to “Relative Slowdown” and changed the caption of Figure 2 to: *“Relative slowdown in mean throughputs between Opacus per-example clipping and the non-private baseline using one A100 GPU. It is defined as private-throughput/non-private-throughput, this means that lower is better. It shows how many times private training is more expensive with a relative slowdown of 1 indicating that Opacus is as fast as non-private training.”*
>
>
> >In lines 399 to 400, could you elaborate on why larger models are too expensive (to have benefit from lower precision)?
>
> Thanks for the question! We clarified this point in the respective discussion of Figure 5: *”Models that are too small do not gain much from TF32, and the larger ones have too little maximum physical batch size to benefit (See detailed discussion of this in Appendix C).”*
>
>
> In Appendix C, we write: *’The speedup observed in Figure 5 peaks at the ”base” model. We believe that the reasons are the following: Speed-ups resulting from TF32 can significantly vary on per case basis as “all storage in memory and other operations remain completely in FP32, only convolutions and matrix-multiplications convert their inputs to TF32 right before multiplication.” (Stosic & Micikevicius, 2021). Until now, TF32 precision benchmarks have been limited to non-DP applications which was one of the reasons we wanted to discuss our observations in DP context. It appears the effectiveness of TF32 arithmetic peaks at “base” configuration. This due to a mix of reasons which are difficult to quantify exactly. Firstly, it is likely that matrix multiplication kernel dominance peaks at this configuration i.e. we have the most parameters whilst the batch size dimension also remains sufficiently large. With large and huge model variants the parameter count still increases but at the cost having very small batch dimension of 10 and 3, respectively. Secondly, we observe similar trend in Figure 2(a) where the discrepancy between dp and non-dp grows as model size gets bigger. This suggests that the dominance of DP operations also grows with the model size. None of the DP-operations are cast as matrix-multiplications and hence won’t benefit from TF32.’*
>
>
> > In the JAX paragraph of section 2.2, could you elaborate on how masking achieves poisson sampling?
>
> Based on your feedback and the feedback of other reviewers we improved the writing of Section 2 and 3 and added a short paragraph about our proposed method. We agree with you that this was not written clearly enough in the first version of the paper.

---

> > ### Comment · Reviewer_L1Bo · 2024-11-24
> >
> > I thank the authors for their response, and believe my comments have been answered.

---

> > > ### Author Response · Authors · 2024-11-24
> > >
> > > Thanks for reading and acknowledging our rebuttal!

---

### Official Review · Reviewer_srbv · 2024-11-03

**Soundness:** 2
**Presentation:** 3
**Contribution:** 2
**Rating:** 5
**Confidence:** 4

**Summary:**

The paper provide a implementation of DP-SGD algorithms with Poisson sampling, and benchmarked a few clipping algorithms in Opacus and jax. The benchmark shows jax is typically faster and ghost clipping algorithms have comparable (in the sense that they are not 10x slower) throughput with non-dp conterparts.

**Strengths:**

The paper provided efficient implementations of DP-SGD with Poisson sampling which would be useful to the community. Also the paper benchmarked the implementation with Opacus and jax with informative results on what clipping algorithms one should choose.

**Weaknesses:**

My major concern is the contribution seems not significant enough, the benchmarks are nice but are not comprehensive enough if viewing it as a benchmark paper. For example, torch compile is not considered in the paper which is one of the most useful emerging acceleration technique in pytorch newer versions and . Also when comparing with Opacus, the backend used by Opacus is not mentioned, Opacus have a few backend methods for efficient clipping, like functorch based implementation and backward hook used approach. More details at the link: https://github.com/pytorch/opacus/tree/main/opacus/grad_sample.

The Poisson sampling implementation is useful I am not sure it is significant enough given the existing tools like Opacus and jax already have pretty good support to non-Poisson sampling, the Poisson sampling should be just changing the data batching parts while other parts are utilizing the existing tools. In addition, the paper did not benchmark the performance difference between non-Poisson sampling and Poisson sampling which is nice to have since Poisson sampling is the main focus of the paper.

**Questions:**

1. Could the authors provide some explanations on the novelty or difficulty in implementing the Poisson sampling?
2. Opacus backend does have optimized implementation of gradient clipping and ghost clipping is now supported as well, and torch compile should be able to improve speed and memory consumption in some cases. I wonder why authors did not compare with the optimizations that can be done in pytorch when comparing with jax.

---

> ### Author Response · Authors · 2024-11-21
> **Answer to Reviewer srbv**
>
> > Could the authors provide some explanations on the novelty or difficulty in implementing the Poisson sampling?
>
> Poisson subsampling results in varying (logical) batch sizes at every step of DP-SGD. This is especially challenging for the JAX implementations as changing tensor sizes (can) trigger time-consuming recompilations. We believe this is the reason why prior JAX implementations do not correctly implement the Poisson subsampling but instead use other sampling methods while using the privacy accounting for Poisson subsampling. Concurrent work that will be published at NeurIPS [1] has also acknowledged issues with JAX recompiling due to varying batch sizes and proposed a different but non-trivial solution.
>
> > Opacus backend does have optimized implementation of gradient clipping and ghost clipping is now supported as well, and torch compile should be able to improve speed and memory consumption in some cases.
>
> We would like to thank the reviewer for the question. In our original submission we briefly (but not prominently enough) mention that we tried
> - We tried ghost clipping but we ran into issues (Section 3). We made this now more prominent by mentioning it in Table 1 with an “?” and in the caption: *”Benchmarked DP-SGD frameworks and libraries. Note that Opacus implements Ghost Clipping but in our experiments the loss does not decrease, thus indicating a problem”*.
> During the rebuttal period we installed the most recent version of Opacus from the git repository but still observe that the model is not learning when training with Ghost Clipping. In fact, while Ghost Clipping was released in version 1.5 of opacus [2] it seems that the maintainers are still resolving issues regarding it [3]. We believe that the findings regarding the throughput of Ghost Clipping will transfer between the implementation of the algorithm in the libraries that we used and Opacus and as the underlying algorithm is the same.
> - we also tried the new torch compile but did not find any benefits with opacus (Section 6) as the speedups are minimal and the compilation time “eats” up these benefits while changing tensor sizes (e.g., through changes in the logical batch sizes) are also retriggering recompilations as in the naive JAX implementation. We will post throughput numbers ASAP.
>
> > Also when comparing with Opacus, the backend used by Opacus is not mentioned, Opacus have a few backend methods for efficient clipping, like functorch based implementation and backward hook used approach
>
> Thanks for raising this point, we looked into this more closely and compared the throughput of all available grad_sample models (“hooks”, “functorch”, “ExpandedWeights” and “GhostClipping”). Functorch yields the same throughput as hooks (our original mode) while expanded weights crashes and GhostClipping does learn anything (see above). We added these details to the paper in Section 3 and Appendix A.3:
>
> *”We benchmark a native PyTorch (Ansel et al., 2024) implementation with PyTorch based libraries Opacus (Yousefpour et al., 2021) (details on gradsampling in Appendix A.3), PrivateVision (Bu et al., 2022), and FastDP (Bu et al., 2023), see Table 1.”*
>
> In Appendix A.3: *”Opacus supports multiple different gradient sampling methods as indicated in the documentation [4]. In our original experiments we used the grad sample mode hooks that is the default. This will use custom opacus modules when they are defined for that layer and functorch as a fallback. Based on the feedback by a reviewer we tried out different methods listed in the documentation for both ResNet and ViT models:
> - $functorch$: We forced opacus to use functorch but did not observe any significant speed differences to using $hooks$. This is in line with the opacus documentation which writes that the speed is $0-50$% slower than $hooks$
> - $ExpandedWeigths$: We tried using this approach but ran into runtime errors. Interestingly, when looking through the issues others have reported issues[5,6] but it seems to be more a PyTorch problem and has not been addressed for years. According to the documentation $ExpandedWeights$ is still in beta status.
> - $GhostClipping$: Note that this method only works for ViT as described in Section 5.1. We did not manage to decrease the loss with this implementation due to the implementation in opacus being unstable but think that the speedups should be similar as observed in our experiments in Section 5.1 as the underlying algorithm is the same.”*
>
> [1] Chua, L., Ghazi, B., Kamath, P., Kumar, R., Manurangsi, P., Sinha, A., & Zhang, C. Scalable DP-SGD: Shuffling vs. Poisson Subsampling. NeurIPS 2024
>
> [2] Opacus v1.5 https://github.com/pytorch/opacus/releases/tag/v1.5
>
> [3] Some bug fix in the implementation https://github.com/pytorch/opacus/pull/664
>
> [4] https://github.com/pytorch/opacus/tree/61ae0ea4fb37a835e93040b5de19e8dfcd465a07/opacus/grad_sample
>
> [5] https://github.com/pytorch/opacus/issues/584
>
> [6] https://github.com/pytorch/opacus/issues/464

---

> > ### Author Response · Authors · 2024-11-27
> > **On PyTorch compilation**
> >
> > > For example, torch compile is not considered in the paper which is one of the most useful emerging acceleration technique in pytorch newer versions and .
> >
> > Thanks for the comment and we wanted to expand on it. We included the experiments for ViT Base on A100 in Figure A.3 in the Appendix and we refer to it in the corresponding paragraph in Section 6. The caption says:
> > “Torch compilation experiments on A100, using the maximum physical batch size for each mode and ViT Base. PyTorch 2 enables compiling the model to (potentially) gain further speed-ups. We tried PyTorch 2 compilation to make a fair comparison with the JAX compilation but did not observe any benefits from it. We found that when trying to compile PyTorch, it first tries to compile but then falls back to NVIDIA kernels and optimizations. In the end, it does not compile, and the throughput is the same. If we take into account the first iteration (w compilation time), it is worse because of the time PyTorch spends trying to compile before falling back to NVIDIA kernels and optimizations. Disregarding the time where PyTorch tries to compile (wout compilation time), leads to nearly the same throughput as the version that does not attempt using PyTorch 2 compiling in the first place.”
> >
> > Note that the changes related to your questions are highlighted in blue as suggested by reviewer ZyeT. Thanks again for providing very valuable feedback.

---

### Official Review · Reviewer_gRw1 · 2024-11-04

**Soundness:** 3
**Presentation:** 1
**Contribution:** 3
**Rating:** 6
**Confidence:** 3

**Summary:**

The paper discusses which aspects influence efficiency and scalability when implementing differentially private stochastic gradient descent (DP-SGD).
The authors perform an empirical study to identify which operations hinder computational efficiency and propose an efficient JAX implementation.
By measuring throughput (i.e., how many samples are processed per unit of time during training), the authors compare their implementation with existing library implementations of DP-SGD, as well as with non-DP SGD implementations.
With their experiments, the authors find that they can almost match the throughput of a PyTorch implementation of SGD.

**Strengths:**

The paper tackles an important problem in privacy-preserving ML, as DP-SGD is often regarded as the most commonly used approach to adopt DP in practice.
The authors identify the main sources of inefficiency, namely Poisson subsampling, gradient clipping, and the per-sample gradient computation that characterize DP-SGD.
With respect to Poisson subsampling, the authors note that DP-SGD libraries which do not implement it, may have weaker privacy guarantees than claimed.
With respect to gradient clipping, the authors empirically compare standard per-example clipping with more recent optimizations, i.e., ghost clipping and book keeping.
The empirical evaluation is extensive and focuses on vision classifiers (ViT Base and ResNet).
While the authors do not introduce novel techniques, their analysis of existing strategies (and their re-implementation) promises to be very useful in practical settings.

**Weaknesses:**

**Summary of "lessons learnt"**

While the empirical evaluation proposed by the authors can be very useful to identify which strategies are more effective when implementing DP-SGD efficiently, the paper is (in this sense) difficult to parse.
To expand on this, I find that absence of a clear "lessons learnt" section/table/summary/discussion can negatively affect the impact of the contribution.
By the end of the paper it is not easy to remember which techniques have the biggest impact on efficiency.
The conclusions section, for instance, simply reminds the reader that feasible speed-ups include "efficient implementations of DP-SGD, JAX, efficient implementation of Poisson sampling, lower precision using TF32".
From a practitioner's perspective, an informative summary is therefore missing and would significantly increase the value of the contribution, in my opinion.

**Structure**

With reference to my previous comment, the structure of the paper does not help readability.
The main result (figure 1) is presented in the introduction, where most of the information necessary to parse the visualization has not been presented yet.
The Experiment Overview in Section 3 does not effectively present an overview of the experiments, as it is not clear which libraries will be compared, with which clipping strategies, etc.


**Throughput as the only quality measure**

While I do agree that throughput is an intuitive measure to quantify efficiency, I think that some more extensive discussion on how the various strategies to increase efficiency affect model utility is missing.
As there are many moving parts contributing to the final result (i.e., a model trained with DP-SGD), it would in my opinion be important to discuss how to improve throughput while maintaining similar model utility.
To be more specific, it is mentioned that Poisson subsampling is sometimes not implemented, as efficient implementations are difficult.
I can expect this to impact not only efficiency, but also model utility (in terms of, e.g., classification accuracy) and privacy.
In this sense, a comparison of the throughput between your implementation which uses Poisson sampling and an implementation which foregoes Poisson sampling is, I think, unfair.
On the one hand, your implementation may results to be "less efficient" in this case, as it may process instance more slowly.
On the other hand it is unclear what the impact on privacy and utility is.
While you do specify in section 2.1 that all the experiments are "seeded and have the same logical batch size", it is not clear whether all experiments obtain models which have comparable utility and privacy.
Incidentally, I would expect this detail on how seeding is used to be part of the experiments section, rather than the background.

On a similar note, your claim that (see, e.g., contribution 2 in section 1) DP-SGD "costs between 2.6 and 3.2 times more that non-private training" may be misunderstood.
It may be unclear to the reather whether this suggest that the overall cost of the training procedure has a factor ~3 difference (e.g., until model convergence), or whether this refers to, e.g., a single epoch.
In any case, it is necessary to explicitly discuss how the performance of a model trained with SGD and with the various DP-SGD approaches compare.


**Importance of Poisson subsampling and of lower precision**

You mention that foregoing a proper implementation of Poisson subsampling, as well as using lower precision, can both improve efficiency at the possible cost of privacy (and utility?).
I think that more insights on this could be within the scope of your investigation.
You mention that TF32 can increase throughput, but if this happens at a huge privacy cost, a practitioner would likely not adopt this strategy.
While you mention that this may be a topic for further research, I do think that some more insights on this would be important to actually understand the impact of TF32 on private model training, on a similar line of thought as discussed in the previous section.
I have similar questions with respect to Poisson subsampling, where the impact on privacy is not clearly quantified.

**Questions:**

In addition to the points discussed in the weaknesses section, I have these further questions.

* In your experiments you use a privacy budget of $\epsilon=8$, which is arguably on the upper end of what is usually understood to be a "private model". How does this choice impact the outcome of your experiments?
* In Table A3, you compare non-private models with optimal DP hyperparameters against DP models. Why? How is this a useful or fair comparison?
* What makes it difficult to implement Poisson subsampling?
* While you say that Poisson subsampling is often not properly implemented, I understand that all the methods you compare against do in fact implement it. Is this the case? And if yes, why do you not compare against methods which do not implement Poisson subsampling?

---

> ### Author Response · Authors · 2024-11-21
> **Answer to reviewer gRw1 (1/2)**
>
> > Summary of "lessons learnt"
>
> Thanks for this suggestion! We added Table 4 to the Section 8 (Conclusion) with a brief summary of the lessons learnt.
>
> > whether this refers to, e.g., a single epoch
>
> Thanks for the suggestion, we changed this in the contributions: *“We find that non-optimized training with DP-SGD costs per-epoch between 2.6 and 3.2 times more than non-private training for ViT and 4 to 8 times for ResNets (See Section 4).”*
>
> > In your experiments you use a privacy budget of eps=8, which is arguably on the upper end of what is usually understood to be a "private model". How does this choice impact the outcome of your experiments?
>
> In our experiments, we only measure the throughput which we measure as processed examples per second and only use the test accuracy as a metric for validating that all approaches are implemented correctly. We agree that decreasing the epsilon will most likely have an effect on the optimal hyperparameters in terms of test accuracy but in terms of throughput only the (logical) batch size will have an effect on the throughput as it changes the number of steps per epoch but it is likely that the change is small [1,2].
>
> The other hyperparameters like $learning rate$, $clipping bound$ and $noise multiplier$ will not have an effect as the computational cost does not change when modifying the values used for them. The $number of epochs$ won’t change the throughput as we divide the time by the number of processed samples.
>
> > In Table A3, you compare non-private models with optimal DP hyperparameters against DP models. Why? How is this a useful or fair comparison?
>
> We agree with the reviewer that the optimal hyperparameters for non-DP training would differ from the optimal hyperparameters with DP but we focus on the computational cost of DP deep learning and want to showcase the cost of the additional operations done under DP regardless of the optimal hyperparameters that could be very problem specific.
>
> One illustration of the difference between the optimality of hyperparameters can be found in concurrent work that will be published at NeurIPS [4] in their Figure 3 that showcases that large batch sizes with Poisson subsampling is not optimal for non-DP training in comparison to Poisson subsampling with large batch sizes being optimal for DP training.
>
> > What makes it difficult to implement Poisson subsampling?
>
> Poisson subsampling results in varying (logical) batch sizes at every step of DP-SGD. This is especially challenging for the JAX implementations as changing tensor sizes (can) trigger recompilations. We believe this is the reason why prior JAX implementations do not correctly implement the Poisson subsampling but instead use other sampling methods while using the privacy accounting for Poisson subsampling. Concurrent work that will be published at NeurIPS [4] has also acknowledged issues with JAX recompiling due to varying batch sizes and proposed a different but less efficient solution for our setting.
>
> > While you say that Poisson subsampling is often not properly implemented, I understand that all the methods you compare against do in fact implement it. Is this the case? And if yes, why do you not compare against methods which do not implement Poisson subsampling?
>
> We would like to clarify that none of the implementations but Opacus correctly implements Poisson subsampling and that we add the Poisson subsampling to these to allow for a fair comparison of the throughput and maximum possible physical batch size. Furthermore, we propose the masked DP-SGD algorithm as an alternative to the naive implementation of Poisson subsampling in JAX that is prone to recompilations.
> We do not compare to implementations that don’t use Poisson subsampling as very recent work (NeurIPS 2024) [4] shows that Poisson subsampling is crucial in achieving the optimal utility under DP (Figure 3, left) when all sampling mechanism are applied correctly while other sampling mechanisms can be beneficial in the non-DP setting (Figure 3, right).
> Furthermore, [3,4] ([3] being from ICML 2024) have shown that basing the accounting on Poisson subsampling while implementing a different sampling strategy can lead to a significant difference in the actual privacy loss compared to the accounted one. Especially noteworthy is Figure 3 (middle) of [4] which illustrates the required $noise_multiplier$ for the different sampling mechanisms with the Poisson subsampling $noisemultiplier$ being constantly lower than the others. Another very recent paper [5] (preprint got published on 15 November) also showcases this problem in their Figure 1.
> We already cited the ICML paper [3] but we will add a citation to the NeurIPS paper [4] and the arXiv preprint [5] in the revision of the paper.

---

> > ### Author Response · Authors · 2024-11-21
> > **Answer to reviewer gRw1 (2/2)**
> >
> > > Importance of Poisson subsampling and of lower precision
> >
> > Thanks, for the question. Regarding the Poisson subsampling, we added citations to the recent papers [3,4,5] that we discussed above. These works suggest that the privacy implications of incorrectly implementing the Poisson subsampling are problem dependent and indicate that it is not possible to make a universal statement. Regarding lower precision there are many works trying to understand the impact of floating point precision on DP guarantees such as the one we already cited [6] in Section 5.2. We will cite further works [7,8,9] but we are unaware of any work that specifically focuses on the TF32 precision.
> >
> > [1] Räisä, O., Jälkö, J., & Honkela, A. Subsampling is not Magic: Why Large Batch Sizes Work for Differentially Private Stochastic Optimisation. ICML 2024.
> >
> > [2] Ponomareva, N., Hazimeh, H., Kurakin, A., Xu, Z., Denison, C., McMahan, H. B., ... & Thakurta, A. G. (2023). How to dp-fy ml: A practical guide to machine learning with differential privacy. JAIR, 77, 1113-1201.
> >
> > [3] Chua, L., Ghazi, B., Kamath, P., Kumar, R., Manurangsi, P., Sinha, A., & Zhang, C. How Private are DP-SGD Implementations?. ICML 2024
> >
> > [4] Chua, L., Ghazi, B., Kamath, P., Kumar, R., Manurangsi, P., Sinha, A., & Zhang, C. Scalable DP-SGD: Shuffling vs. Poisson Subsampling. NeurIPS 2024
> >
> > [5] Annamalai, MSMS, Balle, B, De Cristofaro, E, Hayes, J. To Shuffle or not to Shuffle: Auditing DP-SGD with Shuffling. arXiv:2411.10614
> >
> > [6] Mironov, I. (2012, October). On significance of the least significant bits for differential privacy. In Proceedings of the 2012 ACM conference on Computer and communications security (pp. 650-661).
> >
> > [7] Jin, J., McMurtry, E., Rubinstein, B. I., & Ohrimenko, O. (2022, May). Are we there yet? timing and floating-point attacks on differential privacy systems. In 2022 IEEE Symposium on Security and Privacy (SP) (pp. 473-488). IEEE.
> >
> > [8] Casacuberta, S., Shoemate, M., Vadhan, S., & Wagaman, C. (2022, November). Widespread underestimation of sensitivity in differentially private libraries and how to fix it. In Proceedings of the 2022 ACM SIGSAC Conference on Computer and Communications Security (pp. 471-484).
> >
> > [9] Jin, J., Ohrimenko, O., & Rubinstein, B. I. (2022). Getting a-round guarantees: Floating-point attacks on certified robustness. arXiv preprint arXiv:2205.10159.

---

> > > ### Comment · Reviewer_gRw1 · 2024-11-26
> > >
> > > Thank you very much for your rebuttal, and in particular for the better clarity on whether the approaches you compare to do/do not implement Poisson sampling.
> > > However, I still think presentation should be improved upon.
> > > For instance, the questions raised by reviewer ZyeT seem to me to be still unanswered.
> > > I do understand that measuring throughput is the most straightforward way to measure efficiency and that the negative impact on privacy of, e.g., not using Poisson sampling or using TF32 precision may be very problem specific.
> > > But precisely for this reason, I feel like some results in this direction are missing and would be helpful.
> > > To be specific, the question mark in table 5 for TF32 makes the comparison difficult to judge, as for your setting TF32 may/may not offer the same privacy from a practical perspective.
> > >
> > > I thank you again for the detailed answers but I will for now keep my score.

---

> > > > ### Author Response · Authors · 2024-11-26
> > > > **Answer to Reviewer gRw1**
> > > >
> > > > > Thank you very much for your rebuttal
> > > >
> > > > Thanks for reading our rebuttal and acknowledging it.
> > > >
> > > > > For instance, the questions raised by reviewer ZyeT seem to me to be still unanswered.
> > > >
> > > > We would like to point out that we just replied to reviewer Zyet again. This took some time as we were adding experiments for all precision modes with JAX and compared to the truncated Poisson subsampling in [1] (NeurIPS 2024 paper).
> > > >
> > > > > To be specific, the question mark in table 5 for TF32 makes the comparison difficult to judge, as for your setting TF32 may/may not offer the same privacy from a practical perspective.
> > > >
> > > > We agree with you that the practical implications of TF32 are not clear but after conducting an extensive literature review we were unable to find any related work that could provide insights in that direction. We added the question mark to the Table 5 to make clear that there **might** be additional privacy issues but we would like to keep a thorough investigation of the practical privacy implication for future work.

---

### Author Response · Authors · 2024-11-21
**General response to the reviewers**

We would like to thank the reviewers for their valuable feedback that we incorporated into a new revision of the paper. We also included the updated numbers using the same HPC environment for our Masked DP-SGD implementation. Note that the numbers only changed very slightly and that the findings and conclusions don't change.

We posted answers to individual questions below the respective reviews.

---

> ### Author Response · Authors · 2024-11-21
> **Summary of main changes in revision**
>
> These are the main changes in the revision. We added the reviewers that requested the changes.
>
> - Changed Figure 2 axis label (L1Bo)
> - Restructured Sections 2 and 3 and adding Table 1(gRw1, ZyeT)
> - Added more details on our method masked DP-SGD (ZyeT, L1Bo)
> - Added Table 4 with lessons learnt to Conclusion (reviewer gRw1)
> - Cited very recent works on Poisson subsampling and privacy implications [1,2] (Support general discussions with most reviewers.)
> - Appendix C and main text discussion of benefit of TF-32 for models of different sizes (L1Bo)
> - Appendix A.3 and main text discussion of other clipping modes in Opacus (srbv)
>
> [1] Chua, L., Ghazi, B., Kamath, P., Kumar, R., Manurangsi, P., Sinha, A., & Zhang, C. Scalable DP-SGD: Shuffling vs. Poisson Subsampling. NeurIPS 2024
>
> [2] Annamalai, MSMS, Balle, B, De Cristofaro, E, Hayes, J. To Shuffle or not to Shuffle: Auditing DP-SGD with Shuffling. arXiv:2411.10614

---

> > ### Author Response · Authors · 2024-11-26
> > **Summary of changes after comments from the reviewers**
> >
> > We would like to highlight the following changes in the revision based on the interaction with reviewer ZyeT:
> >
> > - We compare our masked DP-SGD to current work [1] (NeurIPS 2025 paper) in Appendix D and find that for our setting it is more computationally efficient in terms of computed gradients.
> > - We highlighted all changes in the text in comparison to the original submission in blue.
> > - Included JAX results with FP32 and TF32 and clarified when TF32 is used
> > - We revised Tables 1 to include citations to the methods there as well
> > - Fixed a typo where we called Opacus “o-flat” in Figures 3 and 7

---

> ### Author Response · Authors · 2024-11-28
> **Regarding the utility/privacy trade-off (Table A2)**
>
> Reviewer ZyeT brought to our attention that there might be some unclarity about the privacy/utility trade-off of the benchmarked methods and we want to clarify that the optimized algorithms do not have any impact on the trade-off unless the precision changes.
>
> We would like to point the reviewer(s) to Table A2 in the Appendix where we have updated the utility numbers for fine-tuning a ViT Base on CIFAR-100 using all benchmarked approaches and showcase that there is no significant difference between the benchmarked methods. We additionally added pointers to the Table A2 in multiple places in the paper (Section 5.2 and 6) to make this point more clear.
>
> All of the methods achieve a test accuracy of ~82% under $\epsilon=8$ and $\delta=1e-5$ when using FP32. This is expected as the optimizations do not change anything theoretical about the operations that could alter the utility/privacy trade-off but make the operations more computationally efficient. (The tiny differences might stem from non-deterministic changes due to the change of operations or CUDA optimizations).
>
> In our experiments changing to TF32 did not change the accuracy in comparison to using FP32 which has also been observed in more comprehensive benchmarks in the non-private application [15].
>
>
> **Details on the change of numbers**
>
> All approaches now use the exactly same weights `vit-base-patch16-224-in21k` (PyTorch previously used a slightly better pre-training as the weights were `vit_base_patch16_224.augreg_in21k` but these seem to be unavailable in JAX) and furthermore all approaches zero the weights of the classification layer (We missed this earlier). Note that all of these changes do not affect the throughput or maximum physical batch size at all but only the utility.
>
> **Why is no change in utility/privacy expected**
>
> This is expected as the optimizations do not change anything theoretical about the operations that could alter the utility/privacy trade-off but make the operations more computationally efficient. One of the papers that we benchmark [14] also highlights this in their abstract where they write: *“We propose an efficient and scalable implementation of this clipping on convolutional layers, termed as the mixed ghost clipping, that significantly eases the private training in terms of both time and space complexities, without affecting the accuracy.”*
>
> Our own method masked DP-SGD builds on-top of the same operations as typical DP-SGD but avoids recompilations. There is no change in terms of the clipping operation, noise addition or sampling but our proposed method enables the use of JAX for DP-SGD using Poisson subsampling and is more efficient than concurrent work that will be published at NeurIPS 2024 [4] as discussed with reviewer ZyeT and showcase in Appendix D.
>
> **What is the difference between using Poisson subsampling and not Poisson subsampling in terms of utility/privacy trade-off**
>
> We cited multiple papers [3,4,5] in the revision that discuss the importance of Poisson subsampling for achieving the optimal privacy/utility trade-off.
>
> [3] Chua, L., Ghazi, B., Kamath, P., Kumar, R., Manurangsi, P., Sinha, A., & Zhang, C. How Private are DP-SGD Implementations?. ICML 2024
>
> [4] Chua, L., Ghazi, B., Kamath, P., Kumar, R., Manurangsi, P., Sinha, A., & Zhang, C. Scalable DP-SGD: Shuffling vs. Poisson Subsampling. NeurIPS 2024
>
> [5] Annamalai, MSMS, Balle, B, De Cristofaro, E, Hayes, J. To Shuffle or not to Shuffle: Auditing DP-SGD with Shuffling. arXiv:2411.10614
>
> [14] Bu, Z., Mao, J., & Xu, S. Scalable and efficient training of large convolutional neural networks with differential privacy. NeurIPS 2022.
>
> [15] Dusan Stosic and Paulius Micikevicius. Accelerating AI training with NVIDIA TF32 tensor cores. https://developer.nvidia.com/blog/accelerating-ai-training-with-tf32-tensor-cores/, 2021

---

### Meta-Review · Area_Chair_cLwo · 2024-12-13

**Metareview:**

The paper investigates the computational cost and efficiency of DP-SGD training. Some concerns of the reviewers have not been addressed properly. The reviewer pointed out that the dynamic batch size is the major technical difficulty for implementing Possion sampling and they think that this should be highlighted and approaches to solve this problem should be discussed. Moreover, the reviewer also has some concerns on the impact of the proposed method, as it provides limited insight into current DP implementations with Poisson subsampling and a benchmark is not comprehensive enough. Based on the reviews and discussions, the paper could not get accepted this time. Please address the reviewers' concerns properly and resubmit the paper to the next publication venue.

**Additional Comments On Reviewer Discussion:**

- Dynamic batch size is the major technical difficulty for implementing Possion sampling
- Impact of the proposed method, as it provides limited insight into current DP implementations with Poisson subsampling
- A benchmark is not comprehensive enough

---

### Decision · Program_Chairs · 2025-01-22

Reject